# Chemosensory Receptors in Vertebrates: Structure and Computational Modeling Insights

**DOI:** 10.3390/ijms26146605

**Published:** 2025-07-10

**Authors:** Aurore Lamy, Rajesh Durairaj, Patrick Pageat

**Affiliations:** 1Department of Bioinformatics and Chemical Communication (D-BICC), Research Institute in Semiochemistry and Applied Ethology (IRSEA), 84400 Apt, France; a.lamy@irsea-institute.com (A.L.); r.durairaj@irsea-institute.com (R.D.); 2Department of Chemical Ecology (D-EC), Research Institute in Semiochemistry and Ethology (IRSEA), 84400 Apt, France

**Keywords:** chemosensory receptors, signaling, modeling, GPCR, chemical communication, computational tools

## Abstract

Chemical communication is based on the release of chemical cues, including odorants, tastants and semiochemicals, which can be perceived by animals and trigger physiological and behavioral responses. These compounds exhibit a wide size and properties range, spanning from small volatile molecules to soluble proteins, and are perceived by various chemosensory receptors (CRs). The structure of these receptors is very well conserved across all organisms and within the family to which they belong, the G-protein-coupled receptor (GPCR) family. It is characterized by highly conserved seven-transmembrane (7TM) α-helices. However, the characteristics of these proteins and the methods used to study their structures are limiting factors for resolving their structures. Due to the importance of CRs—especially olfactory and taste receptors, responsible for two of our five basic senses—alternative methods are utilized to overcome these structural challenges. Indeed, in silico structural biology is an expanding field that is very useful for CR structural studies. Since the 1960s, many algorithms have been developed and improved in an attempt to resolve protein structure. We review the current knowledge regarding different vertebrate CRs in this study, with an emphasis on the in silico structural methods employed to improve our understanding of CR structures.

## 1. Introduction

All animals perceive different signals from their environments through their senses, which provide important information that influences their behavior, such as foraging or seeking mates. They can also use these cues to communicate with other individuals [1,2,3,4,5,6].

These signals have different properties; thus, they are perceived by specific sensory receptors that are able to convert the environmental data into neural signals. Herein, we focus on the chemical cues that are part of these signals, including odorants, tastants, and semiochemicals. Odorants are volatile chemicals weighing less than 300 g/mol that are detected by the olfactory system (OS), the Grueneberg ganglion (GG), and the septal organ of Masera (SOM) [7,8]. Tastants are chemical compounds perceived by the gustatory system (GS) and are identified by five basic tastes: sweet, sour, salty, bitter, and umami. Lastly, semiochemicals are substances secreted by one individual and perceived by another individual [3,9]; they are mainly detected by the vomeronasal system (VNS), but also by the OS, the GG and the SOM [7,10,11]. Notably, these three systems are interconnected. For example, the opening of the vomeronasal organ (VNO) can depend on the signals detected by the OS. Moreover, the distinction between the three classes of chemicals is blurred because the same chemical can activate different chemosensory receptors (CRs) [12].

Chemical signals are perceived by CRs in the nasal cavity through the main olfactory epithelium (MOE), the VNO, the GG, or the SOM and in the buccal cavity through the different papillae in vertebrates (Figure 1) [13]. Perception is initiated by the activation of a cell surface CR, which induces the depolarization of the sensory neuron through ionic flux directly or via a signaling cascade.

CRs are seven-transmembrane (7TM) α-helix proteins belonging to the G-protein-coupled receptor (GPCR) family. One of the problems in studying membrane proteins is that they are insoluble in water, making it difficult to study their structures by means of X-ray crystallography or nuclear magnetic resonance (NMR) without altering their conformations [14]. However, important improvements in single-particle cryogenic electron-microscopy (cryo-EM) since 2013 led to the obtention of membrane protein structure with a resolution under 3.5 Å [15,16]. Cryo-EM rapidly became the principal technique for elucidating membrane protein structures, including CRs [17]. This method has provided the majority of structures for these proteins. However, cryo-EM shows some limitations like the need for a meaningful concentration of protein, which is difficult to obtain with the low expression of CRs [18]. Therefore, in silico studies have been conducted to create three-dimensional (3D) models of the structures of CRs faster and with less resource consumption than experimental methods. Different computational methods have been conceived over the last several decades. These are initially developed by mimicking the biological folding process or designed based on structures that have already been obtained. Many algorithms have since been improved or developed with advances in artificial intelligence (AI), leading to state-of-the-art deep learning models which are capable of predicting structural models for millions of proteins rapidly and with high confidence [19].

## 2. Features of GPCRs

### 2.1. The Classification

The GPCR family is a large family of around 1000 genes in humans [20]. It is the largest group of eukaryotic membrane receptors, which respond to a wide range of external signals, and is highly diverse. These receptors can activate heterotrimeric G-proteins (guanine nucleotide-binding proteins) in the cell. Human G-proteins are divided into four major families according to their G-alpha (Gα) subunit (G_s_, G_i/o_, G_q/11_, and G_12/13_), each with a different signaling cascade, but all involving the catalysis of the guanosine diphosphate (GDP)/guanosine triphosphate (GTP) exchange, leading to subunit dissociation. Mammalian GPCRs can be classified into five families according to their sequence homology and pharmacological features with the ABC classification [21]. The class A rhodopsin-like GPCRs are the biggest family, divided into aminergic, peptide, protein, lipid, melatonin, nucleotide, steroid, dicarboxylic acid, sensory, and orphan subgroups. The class B secretin receptor like family is composed of the secretin (B1) and adhesion (B2) subfamilies. Class C is composed of metabotropic glutamate receptors (mGluRs), γ-aminobutyric acid B (GABA_B_) receptors, Ca^2+^-sensing receptors (CaSRs), taste receptor 1 (T1R), vomeronasal type 2 receptor (V2R), olfactory receptor C in fish, and several orphan receptors [22]. Class C proteins are characterized by their complex structure with multi-domain and dimeric assembly. The class F, or frizzled/smoothened-like, family is composed of frizzled receptors (FZDs) and the smoothened receptor (SMO). Also, class T (or T2 for taste 2) is a recent family that shows structural similarity to class A GPCRs but with low sequence identity (<20%) with previously known types of GPCRs, leading to the creation of this new class of T GPCRs [20,23]. Another classification can be used based on phylogenetic relationships, namely, the GRAFS system, composed of the five families: glutamate, rhodopsin, adhesion, frizzled/taste 2, and secretin receptor families [24].

### 2.2. Roles and Functions

GPCRs are involved in a wide range of biological processes due to their ability to perceive a signal from outside the cell and transmit its information inside the cell. Their roles involve the mediation of sensory information, such as light perception, taste, olfaction, pheromone sensation, the mediation of neuropathic pain, or endocrine regulation [20]. Because of their importance in these biological events, the structure of GPCRs has long been studied for pharmaceutical purposes. Approximately 40% of the drugs approved by the Food and Drug Administration (FDA) in 2020 target GPCRs, the majority of which are class A GPCRs (96%) [25].

### 2.3. The Conserved Structure

Structural studies of all membrane proteins are difficult owing to the limitations of the methods. Indeed, X-ray crystallography requires a crystallized protein, but membrane proteins are too hydrophobic to be easily solubilized in water. For NMR, the problem is the size of the micelles needed [14]. The first GPCR structure was deposited in the Protein Data Bank (PDB) in 2000; thanks to its high concentration, it was bovine rhodopsin obtained by X-ray crystallography (PDB ID: 1F88) at a resolution of 2.8 Å [26]. For seven years, only squid and bovine rhodopsin structures were available in the PDB, until the resolution of the second GPCR structure—namely, the human β2-adrenergic receptor (PDB ID: 2R4R, 2R4S, 2RH1)—in 2007 [27,28].

Of the 226,353 protein structures listed on the PDB website, only 14,297 (6.32%) structures are referred to as membrane proteins as of February 2025. These values reflect the difficulties in obtaining membrane protein structures, since it is estimated that 27% of the human proteome is α-helical transmembrane proteins [29].

All GPCRs share a common structure for the membrane part, with seven transmembrane (TM) helices organized in the same way for each receptor. These TM helices form a helical bundle conformation specific to GPCRs (Figure 2), and it is maintained by conserved between TMs. The N-terminal part varies in its length and role but is always exposed to the outside of the cell, whereas the C-terminal part is a short α-helix exposed to the inside of the cell and involved in the binding of the G-protein. The loops also vary in length and structure, but some similarities have been found in different GPCR classes. Indeed, a disulfide bond is well conserved between extracellular loop 2 (ECL2) and TM3. Moreover, the intracellular loops are conserved because of their role in binding the G-protein [20,30].

## 3. Vertebrate Chemosensory Receptors (CRs)

### 3.1. Olfactory Receptors (ORs)

#### 3.1.1. Genes and Evolution of ORs

The authors of the initial work on the discovery of ORs in 1991 estimated that there were between 100 and 200 genes in the rat genome [31]. However, the whole-genome sequencing of many species allowed the OR gene family to be studied in more detail. The number of OR genes varies greatly between species: mice have more than 1000 genes, but humans have about 350 genes, while fishes have small OR gene families of about 100 genes (Figure 3) [32,33]. Many species have numerous pseudogenes, especially primates [34]. Despite the small number of type I genes, the OR diversity is higher in fishes than in mammals or birds [33].

As their name suggests, ORs are involved in the sense of smell, and this was the first role assigned to them. However, studies have shown that they are expressed in many tissues other than the MOE, such as the liver, lungs, and heart [36]. This wide-ranging expression is explained by their role in a large number of biological processes, for example, the mediation of oxygen homeostasis during breathing, maintaining glucose homeostasis, the modulation of blood pressure, and so on [37].

According to phylogenetic analysis, ORs can be divided into different groups and subgroups. Initially, two classes of ORs were proposed: fish-like class I and mammalian-like class II [38]. Nevertheless, more recent studies conducted on six genomes have led to a new classification of ORs. The first level is types I and II, with type I being the most diversified and type II being found in fishes and amphibians (Figure 4). The divergence between these types seems to have preceded the divergence between jawed and jawless vertebrates. Thus, it is thought that the most recent common ancestor (MRCA) of all vertebrates already had at least two OR genes for types I and II. Then, there are 13 groups: α, b, Υ, δ, ε, ζ, lamp-a, and lamp-b belong to type I, and η, θ1, θ2, κ, and λ belong to type II. The lamprey, which is a jawless vertebrate, possesses its own type I ORs (lamp-a and lamp-b) because evolutionary divergence made them too different to classify as any of the six types I groups. Fish have retained most OR groups, except for group α (which has completely disappeared in fishes) and, to a lesser extent, group Υ (which is present in only a few species, probably because their environment has not changed much from that of the common ancestor). Indeed, group α and Υ genes have undergone a large expansion in the tetrapod lineage due to the acquisition of the ability to detect airborne odorants during terrestrial adaptation. Finally, only the α and Υ groups are conserved in mammals. Surprisingly, the evolution of groups θ1, θ2, κ, and λ differed from that of the other ORs, with rare gene duplications or losses of genes, suggesting that they acquired new functions impacting the survival of organisms [33,38,39,40,41].

Despite the expansion of OR genes in the tetrapod lineage, 41% of human OR genes are pseudogenes (Figure 3) [35]. Two hypotheses have been proposed to explain this high proportion of pseudogenization: on the one hand, the vision priority hypothesis is based on the fact that humans have developed a trichromatic color vision system, and therefore, the requirement for olfaction declined [42]; on the other hand, the brain function hypothesis is based on the idea that a higher level of brain function can improve the ability to smell and process the information [43]. However, analyses seem to refute this type of hypothesis in favor of explanations more specific to each species and their ecological niche [44]. Moreover, pseudogenes are not necessarily nonfunctional; they can be involved in gene regulation through their ribonucleic acid (RNA), and some of them are functional at the protein level, such as OR1E3, which lacks two TMs but retains the excitatory ability in response to ligands [45].

OR genes are distributed throughout the mouse genome, but chromosomes 2, 7, 9, and 11 have the largest number of OR genes [46]. Since ORs represent one of the largest gene families, a classification based on their similarity has been established to name these myriad receptors. ORs are named according to the following model: ORNXM, with family ‘N’ being the number of grouping receptors with at least 40% identity, subfamily ‘X’ being a letter grouping those with at least 60% identity, and ‘M’ corresponding to the number of this receptor in its group. The letter P can be added at the end to indicate a pseudogene [38].

#### 3.1.2. Expression of ORs

As OR expression appears to follow the one-receptor–one-neuron rule [47], many studies have explored the mechanism behind the OR choice in the olfactory sensory neuron (OSN). It appears that lysine-specific demethylase 1 (LSD1) and adenylate cyclase 3 (ACIII) are necessary for the initiation and stabilization of OR expression. Indeed, all OR genes are silenced at an earlier stage of development [48], and LSD1 is responsible for desilencing an OR gene and, thus, initiating its transcription. Then, protein kinase R (PKR)-like endoplasmic reticulum kinase *(*PERK*)*, which is involved in the unfolded protein response (UPR), will detect the expression of a functional OR in the endoplasmic reticulum (ER), leading to the PERK-mediated phosphorylation of eukaryotic translation initiation factor 2α (eIF2α). eIF2α is responsible for the translation of activating transcription factor 5 (ATF5), which in turn activates the transcription of ACIII [49]. In addition, ACIII is involved in the downregulation of LSD1, which leads to the stable OR choice by preventing the desilencing of other OR genes [50].

One study showed that receptor-transporting proteins (RTPs) 1 and 2 are necessary for the translocation of ORs from the ER to the plasma membrane [51]. Indeed, the N-terminus of RTPs allows the exit from the ER, and an 80-residue middle region permits the translocation of ORs from the Golgi apparatus to the plasma membrane. The ORs, for their part, interact with the C-terminus of RTPs. A shorter version of RTP1, called RTP1S, has been identified in olfactory tissues and is more efficient in OR transport [52]. The mechanism of action of RTPs involves the recruitment of chaperone proteins: heat shock protein A6 (HSPA6) and Staufen homolog 2 (STAU2). HSPA6 interacts with the N-terminus of RTP1S prior to the OR interaction, whereas the interaction between STAU2 and RTPs is less understood [53]. Another chaperone involved in the correct folding of ORs but without known interactions with RTPs is heat shock cognate protein 70 testis-enriched (Hsc70t), which was detected in OSNs. This protein is known to be expressed in mature spermatozoa, another cell type expressing ORs [54], and a study showed that Hsc70t is capable of increasing OR expression [55].

#### 3.1.3. Signaling Pathway of ORs

Airflow through the nasal cavity, coming from the orthonasal or retronasal pathway, brings chemicals to the surface of the nasal mucus. Odorant-binding proteins (OBPs) solubilize the chemicals, allowing them to bind to ORs, whereas xenobiotic-metabolizing enzymes (XMEs) transform the chemicals, allowing the modulation of the signal [56]. The activation of ORs involves the displacement of TM6, leading to an intracellular signaling cascade. Indeed, the associated heterotrimeric G_olf_ protein changes its conformation when the OR is activated, resulting in the dissociation of GDP and the binding of GTP to the α subunit of the G-protein. The newly activated Gα_olf_ targets ACIII, which catalyzes the conversion of adenosine triphosphate (ATP) into cyclic adenosine monophosphate (cAMP). Then, the opening of cyclic nucleotide-gated (CNG) channels is caused by an increase in cAMP concentration in the cell. These channels allow the entry of cations, especially calcium ions (Ca^2+^), depolarizing the sensory neuron [31,57]. Ca^2+^ also activates Anoctamin 2 (ANO2), a chloride channel also called TMEM16B, leading to an efflux of chloride ions (Cl^-^), amplifying depolarization [58,59,60]. ANO2 can also hetero-oligomerize with ANO6, leading to an increase in chloride flux [61]. Then, mechanisms not yet fully understood result in the desensitization of OSNs. Hypotheses include the inactivation of ORs via phosphorylation by G-protein-coupled receptor kinase (GRK) or protein kinase A (PKA). PKA-mediated phosphorylation results in OR internalization and the closure of CNG channels due to the elevated concentration of calcium [32].

#### 3.1.4. Structure and Binding Site of ORs

ORs are membrane proteins with a length of around 310 amino acids. They are part of the GPCR family, and thus, they have seven TM helices, allowing their insertion into the membrane. Two well-conserved cysteines, one at the beginning of TM3 and one in the middle of ECL2, are involved in a disulfide bridge. They also possess an eighth intracellular helix, which could serve as an anchoring point to the membrane due to its parallel orientation to the membrane [18]. The specificity of the OR is the length of ECL2 compared to other GPCRs, but if it is longer, it is also less conserved [62].

There are conserved motifs in class A GPCR sequences, including OR sequences, but some motifs are specific to ORs (in bold) (Figure 5) [63]:GN in TM1;**LHxPMYFFLxx**L**S**xxD in TM2;**MAY**D(E)RY**VAICxPLxY** in TM3;**S**Y in TM5;KAFSTCxSH in TM6;**PxL**NPxIY**SLRN** in TM7.

Moreover, an ionic bridge between TM3 and TM6 is also involved in OR activation; it is closed when the receptor is inactive, but when it is in an active conformation, the interaction is disrupted, leading to the displacement of TM6. The residues involved appear to be the arginine of the DRY motif of the TM3 helix, with a residue at the beginning of TM6 forming a hydrogen bond (Figure 6). If the TM6 residue is positively charged and sufficiently large (arginine or lysine), a double ionic bridge will form with the addition of a hydrogen bond between the TM6 residue and the aspartic acid of the DRY motif [64].

Thanks to mutagenesis and in silico studies, some residues have been identified as part of the binding site. This binding site involves the upper parts of TM3, TM5, TM6, and, to a lesser extent, TM7. ECL2 also seems important for the binding site [63]. Indeed, ECL2 is implicated in ligand selection and permits the formation of a hydrophobic pocket for the ligand [62].

### 3.2. Trace Amine-Associated Receptors (TAARs)

#### 3.2.1. Genes and Evolution of TAARs

The first identifications of TAARs in diverse tissues of both humans and rodents were made in 2001 by two teams using degenerate polymerase chain reaction (PCR) based on known receptors for amines, such as catecholamine and serotonin receptors [65,66]. However, their first identification in the MOE, with a potential role as CRs, was in 2006. Indeed, among the 15 TAAR proteins of the mouse, 14 are expressed in the MOE at the same level as ORs, the exception being TAAR1 (Figure 7) [67,68,69]. Moreover, TAARs can detect urine from different species, especially rodent TAAR4, which is specific to carnivore urine. By detecting 2-phenylethylamine in urine, TAAR4 induces avoidance behavior, proving the involvement of TAARs in semiochemical communication [68]. TAARs form a new subfamily of class A, or rhodopsin-like, GPCRs [70,71]. Indeed, they show some common features with class A GPCRs, for example, the D(E)RY motif at the end of TM3, but also some specific features, such as a predictive peptide fingerprint motif found only in TAARs.

There are three distinct TAAR subfamilies: (i) TAAR V, conserved exclusively in amphibians and teleosts, (ii) the lamprey TAAR-like group based on 25 such proteins in the sea lamprey, and (iii) the gnathostome or jawed vertebrate TAARs [72,73]. Notably, teleost fish exhibit a significantly larger repertoire of TAAR genes compared to mammals, suggesting the important diversification of this receptor family in aquatic environments (Figure 7). Only the last subfamily is known to be involved in chemosensory perception.

The names of gnathostome TAARs are given according to the order of the genes on chromosomes, and the orthologs have the same number, while the letters correspond to paralogs [70]. These TAARs are well conserved across species, with some exceptions, such as the independent expansion of the TAAR7 and TAAR8 subfamilies specific to the mouse and rat [74]. Considering the high identity and tight clustering of TAAR genes in each species, it was hypothesized that they evolved recently. However, looking at the moderate to low identity between orthologs, it was concluded that they evolved rapidly [65].

The nine families of gnathostome TAARs originated from the TAAR1 gene undergoing eight duplication events, which occurred before the origin of amniotes for TAAR2 to TAAR5 and before the origin of mammals for the others [70,72]. Three distinct subfamilies can be identified in phylogenetic and functional analyses (TAARs 1–4, TAAR 5, and TAARs 6–9). The orientation of the open reading frames is also significant; human TAARs 1–5 and TAARs 6–9 display inverse orientations, and the same arrangement is found in rodents, except for a block composed of some paralogs of TAAR7 and TAAR8 with the same orientation as the TAAR 1–5 group (Figure 8). The ligands of TAARs are also different; the TAAR 1–4 group is activated by primary amines, and the TAAR 5–9 group by tertiary amines. This classification into two groups is the one preserved today [74]. New putative TAAR families specific to eutherians (E1) and metatherians (M1-M3) have recently been found, and they belong to the TAAR 5–9 group [72].

The TAAR genes of teleosts are distributed on different chromosomes, whereas those of tetrapods are found on a single chromosome and, at least in amniotes, grouped in a single cluster [72]. The six human TAAR genes and three pseudogenes are localized on chromosome 6 in the q23 region [65,66]. TAAR genes are composed of one or two exons [70,75].

#### 3.2.2. Expression of TAARs

The expression of TAARs has been detected in the MOE and, for some of them, in the GG [67,76]. Their expression follows the one-receptor–one-cell rule, which is common for many CRs. Each TAAR is expressed in a small subset of OSNs spread randomly in the nasal cavity and where no other CR is expressed.

Like the ORs, TAARs are expressed monoallelically, and in the case that the first selected allele fails to produce a functional TAAR, another TAAR or the other allele is expressed randomly, but not an OR [77]. However, the first difference from ORs is that the nonfunctional TAAR allele remains expressed in the adult, whereas nonfunctional OR alleles are silenced. These expressed nonfunctional TAARs may have another unknown function, or, unlike ORs, they may remain expressed because they are fewer in number, making them less energetically costly to maintain than to silence. Moreover, OR genes are thought to be initially silenced by epigenetic marks, and their expression comes from desilencing, but TAAR genes are not silenced. According to a study of enhancers of OR and TAAR genes, it is hypothesized that the expression of these two types of genes shares a similar mechanism, but the proteins and transcription factors might be different [78].

#### 3.2.3. Signaling Pathway of TAARs

Being expressed in the same cell type as ORs, TAARs are co-expressed with the same Gα, i.e., Gα_olf_ [67]. Thus, TAAR activation is hypothesized to activate this G-protein, leading to a similar signaling pathway to ORs, involving an increase in cAMP concentration by ACIII. This increase leads to the entry of calcium cations because of the cAMP-dependent opening of the CNG responsible for neuron depolarization.

#### 3.2.4. Structure and Binding Site of TAARs

TAARs share the same structural characteristics as ORs, as both are class A GPCRs with the 7TM structure, with the N-terminal part exposed to the outside of the cell and a length of around 350 amino acids. TAAR protein sequences share residue conservation specific to class A GPCRs (Figure 9), such as the conserved cysteines in TM3 and ECL2 involved in the disulfide bridge or the D(E)RY motif of the ionic lock involved in the activation of the receptor [74,79]. These conservations suggest that they share the same mechanism of activation described in the OR section. However, TAAR sequences show a specific conserved motif in TM7: NSxxNPxxY(H)xxxY(F)xWF [70].

Despite the similarity between TAARs and ORs, they have distinct binding sites. Indeed, comparisons of the structures of OR51E2 and TAARs that bind tertiary amines show differences [18,79]. The binding site of TAARs is closer to that of aminergic receptors. The cavity is formed by TM3, TM5–7, and ECL2, which closes it from the outside (Figure 10). In the binding site, Asp^3.32^ (Ballesteros–Weinstein numbering system [80]) is conserved because it is involved in a salt bridge with the amine functional groups of the ligands [79,81,82]. On the contrary, residues 3.37 and 3.38 vary according to the ligand preference of the receptor and could be involved in the selectivity of the ligand [74] (Figure 9).

### 3.3. Vomeronasal Receptors (VRs)

#### 3.3.1. Genes and Evolution of VRs

Initial work conducted to find VRs in rat VNOs led to the identification of 30 VRs and the hypothesis of a family of around 100 genes (Figure 11), divided into two subfamilies according to their co-expression with protein G [83]. The authors of this study identified only Gα_i2_ VRs, later called V1Rs or type 1, located in the apical layer of the VNO. When they tried to use this work to identify homologous human VRs, the sequences found were pseudogenes with a premature stop codon, except for one sequence expressed in the MOE [84]. Currently, five V1R genes are known to be protein-coding genes in humans [85]. A second study was conducted to identify Gα_o_ VRs in the basal layer of the VNO, later called V2Rs or type 2, in the rat. In this study, the authors estimated that there were around 100 V2R genes, divided into six subfamilies. Then, they searched for similar genes in humans and found 10 V2Rs, but introns made it difficult to identify the full coding sequences and thus to know whether they were functional [86]. In amphibians, V1Rs are not expressed in the VNO but are in the MOE, with the co-expression of Gα_i2_, and some V2Rs are also expressed in the MOE [87,88]. Moreover, the expression of a V2R has been detected in the Grueneberg ganglion, indicating the potential role of the GG in the perception of semiochemicals [89]. In teleosts, the receptors homologous to V1Rs can be called ORAs (olfactory receptors related to class A) [90,91], and those homologous to V2Rs can be designated OlfCs (olfactory C family GPCRs) [92,93]. Indeed, they do not possess a VNO, so the equivalent proteins of VRs are expressed in the MOE [94]. Fish V2Rs seem to have evolved to detect amino acids, which can be important chemical cues for them [95]. The number of V1R and V2R genes varies greatly between lineages: they are particularly well developed in glires but absent in reptiles and birds (Figure 11).

VRs are members of the GPCR family; indeed, they show 7TM helices, and the N-terminus is outside the cell. V1R genes are intronless and localized on chromosomes 6, 7, 13, and 17 in the mouse genome, whereas V2R genes have introns and are distributed on chromosomes 3, 5, 6, 7, 10, 14, 17, and X in the mouse [9,46,83,86,96,97].

V1Rs are subdivided into 18 subfamilies (Figure 12A) in mammals according to phylogenetic analysis [98,99]. The classification was initially based on rodent genomes, and some of these families could be rodent-specific, but there are also cow-specific and human-specific families. However, studies on the substitution rate suggest that the common ancestor of primates, rodents, artiodactyls, and carnivores shared the majority of the families [100]. In fish, there are six genes grouped in pairs: ORA1-ORA2, ORA3-ORA4, and ORA5-ORA6 [91]. Like ORs, V1Rs follow the one-neuron–one-receptor rule, involving a low expression of each receptor in the VNO [83,101]. Using the same criterion as for V1R, at least 40% identity, tetrapod V2Rs are grouped into five classes (Figure 12B): A (subdivided into nine clades), B, C (also known as V2R2), D, and E [102,103]. Classes A, B, D, and E follow the one-receptor–one-neuron rule [104,105]. However, the C class V2R group is composed of seven members in the mouse and is co-expressed with another V2R class; the C class was first called V2R2 [106,107]. In fish, the classification includes 19 fish-specific classes, from 1 to 16 and a1, a2, and a3, where a2 and a3 are cartilaginous fish-specific families. They also share class C, which is co-expressed with the other classes [108,109].

#### 3.3.2. Expression of VRs

Similarly to the expression of ORs, when a functional V1R is expressed, an unknown negative feedback signal leads to the silencing of the other V1R genes. Thus, when a nonfunctional V1R is expressed, another V1R will be co-expressed, but the whole cluster of nonfunctional V1Rs will be silenced, a phenomenon referred to as “cluster lock” [110]. Indeed, it is speculated that a regulatory sequence is present in the cluster that acts on multiple genes, but only one gene at a time in the given cluster [111]. A study based on an OR sequence with a V1R promoter showed that the OR sequence has the ability to provide the V1R negative feedback. This observation suggests that OR and V1R share the same mechanism, but further investigation must be conducted [112].

The expression of V2Rs is hypothesized to be sequential. First, the ABDE family V2R gene is expressed, leading to the expression of the corresponding C family V2R gene [107]. Then, as suggested in one study, the C family V2R forms a heteromeric complex with an ABDE family V2R, which promotes cell surface expression [113].

V1Rs and V2Rs are not transported to the membrane by the same mechanism [114]. The transport mechanism for V1R is unknown but differs from that for V2R and ORs. For V2Rs, the hypothesis is that calreticulin, present in the ER, blocks V2Rs in the ER membrane, but its presence is important because of its protein-folding role in the cell. Thus, calreticulin is downregulated in V2R vomeronasal sensory neurons (VSNs), and a homolog, calreticulin4, which is pseudogenized in humans, maintains the protein-folding ability without blocking V2Rs in the ER membrane. Moreover, calreticulin4 is specifically expressed in the VSNs of species with a large repertoire of V2Rs. Then, for some V2Rs, two proteins form a complex: a member of the major histocompatibility complex (MHC) class Ib H2M10 family and β2-microglobulin (β2m) [115,116]. They seem to both be required for the proper transport of these V2Rs to the plasma membrane and for their stability. They may also be involved in the reception of pheromone ligands as receptors, as modulators, or in the signaling response.

#### 3.3.3. Signaling Pathway of VRs

The signaling pathways in VSNs are not fully understood to date. Even if V1R and V2R are coupled to different G-proteins, only one pathway is identified for both receptors. Indeed, it is the βγ subunit that activates phospholipase C (PLC), leading to the production of diacylglycerol (DAG) and inositol-1,4,5-triphosphate (IP3) [117]. DAG is responsible for the activation of transient receptor potential channel 2 (TRPC2), a channel allowing the influx of Na^+^ and Ca^2+^ cations, resulting in the depolarization of the VSN [118,119]. IP3 triggers the release of the intracellular Ca^2+^ stock, increasing its concentration in the cell, but the effect on depolarization is negligible [120]. Calcium ions activate calcium-sensing chloride channels, such as the sodium–potassium–chloride cotransporter (NKCC) and ANO1, amplifying VSN depolarization [121,122]. An alternative pathway without TRPC2 might exist in basal cells of the VNO, as TRPC2^−/−^ mice remain capable of detecting semiochemical cues [123]. Indeed, one study suggested that NKCC might be necessary and sufficient for VSN activation [124]. Moreover, in another study, the authors hypothesized that Na^+^/H^+^ exchanger regulatory factor-1 (NHERF1), a PDZ (for post-synaptic density-95, disks-large and zonula occludens-1) domain-containing protein, might interact with V1Rs and Gα_i2_, implicating a novel actor in this pathway only for V1Rs [125]. Calcium ions also bind to the Ca^2+^ sensor protein calmodulin (CaM), which will inhibit TRPC2 via negative feedback, resulting in slow adaptation [126]. Short-term adaptation also takes place in VSNs through the slow inactivation of Na^+^ channels [127].

#### 3.3.4. Structure and Binding Site of VRs

Like TAARs and ORs, V1Rs share features of class A GPCRs, with seven TMs comprising around 310 amino acids. In particular, the sequence identity, which is low within TM domains, suggests an internal binding site across the TMs [83]. The low conservation of the binding site allows the binding of diverse classes of ligands among different V1Rs. Some conserved residues have been identified as V1R-specific, but no motif has been identified because of the poor conservation between V1R families [128]:G, M in TM1;C, R between TM2 and TM3;L in TM3;C between TM4 and TM5;M, L in TM5;H, E, A between TM5 and TM6;L, F in TM6;P in TM7.

The particularity of V2Rs is that they possess a long N-terminal part, around 550 residues, with high variability, in addition to the membrane part composed of the seven TMs, which are more conserved than the V1R membrane part. Because of their sequence identity with the metabotropic glutamate receptor (mGluR) and calcium-sensing receptor (CaSR), the N-terminal region is considered involved in ligand recognition [9,86,96,97,129]. Indeed, the N-terminal region is composed of two domains: the cysteine-rich domain (CRD) and the Venus flytrap (VFT) domain (Figure 13). The CRD is composed of nine conserved cysteines involved in four disulfide bonds within the CRD and a fifth bond with the VFT domain. The VFT domain is composed of two N-terminal domain lobes and a hinge region that allows the lobes to close around the ligands, similar to the Venus flytrap plant [130]. Some residues specific to V2Rs were identified in the membrane region (Figure 14) [97]:G in TM1;AN between TM1 and TM2;C between TM2 and TM3;F, LxK, A in TM3;P between TM4 and TM5;GS in TM5;F, LP, ExK between TM5 and TM6;F, V, F in TM6;E, LxS, FxxK, L in TM7.

### 3.4. Formyl Peptide Receptors (FPRs)

#### 3.4.1. Genes and Evolution of FPRs

FPRs were first detected in 1977 in a study aiming to identify the interaction between leukocytes and N-formylmethionyl peptides that are chemoattractants [131]. Indeed, plasma membrane binding sites were identified for each cell using radioligands, and each binding site was hypothesized to be an N-formyl peptide receptor. FPRs are involved in innate immune responses. Thus, they are expressed in neutrophils and myeloid lineage cells, where they recognize formylated peptides released by bacteria or mitochondria during infection or tissue destruction [132]. However, some murine and human FPRs have been detected in VSNs, at the same level of expression as VRs, suggesting a role in chemosensory perception [133,134]. These receptors are part of the class A or rhodopsin-like family of GPCRs [135]. The hypotheses are that FPRs in the VNO can detect the presence of other species, for example, by detecting bacterial flora in feces, or that they can detect unhealthy conspecifics through their perception of FPR agonists in body fluids.

Three members of the FPR family have been identified in humans (FPR1, FPR2, and FPR3) and seven in mice [136], among which five murine FPRs (FPR-rs1, also called FPR3, FPR-rs3, FPR-rs4, FPR-rs6, and FPR-rs7) are expressed in the VNO (Figure 15). In the mouse, one of them, FPR-rs1, is expressed in both the basal layer of the VSN and immune cells, whereas the others—FPR-rs3, -rs4, -rs6, and -rs7—are found exclusively in the VSN and more precisely in the apical layer [133,134]. However, some species do not possess any FPR like the Bovidae or the Aves (Figure 15).

FPR genes have been identified in numerous vertebrate species, indicating that they appeared early in vertebrate evolution. Evolutionary studies showed that an initial duplication resulted in the split of FPR1 and FPR2/3 in the Placentalia lineage [135,136,137]. Then, a differential expansion arose after the divergence of Primates and Rodentia, leading to FPR3 in humans and FPR-rs2, FPR-rs3, FPR-rs4, and FPR-rs5 in mice. This extensive duplication of FPR2 in the Murinae lineage could be related to the acquisition of the new function of chemosensory perception by FPRs, with FPR-rs3 being the first FPR expressed in the VNO. This split between immune and vomeronasal expression seems to have happened at the root of the Eumuroida lineage [138]. Moreover, human FPR3 was found to be less affected by selective constraints during evolution.

FPR genes form a cluster on chromosome 19 in humans and on chromosome 17 in mice, in a region of conserved synteny with human chromosome 19. FPR genes expressed in the VNO are located outside of the ancestral immune FPR cluster and close to VR genes, enabling their control by promoters derived from V1R or V2R promoters. The first FPRs that appeared in the VNO were expressed in the apical layer, under the control of V1R-derived promoters, and then FPR-rs1 was expressed in the basal layer [138].

**Figure 15 ijms-26-06605-f015:**
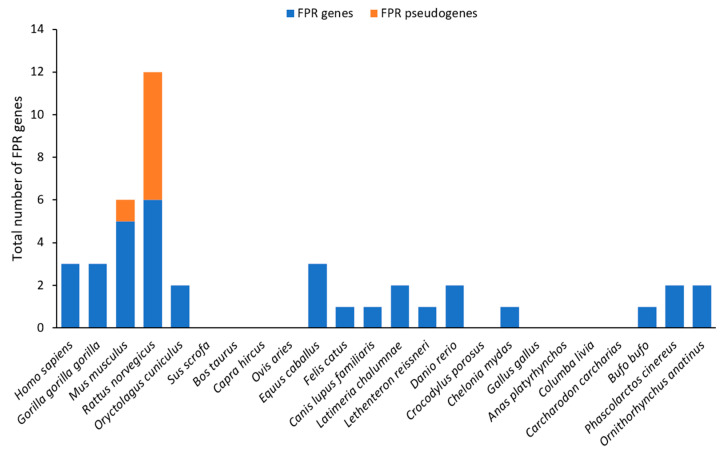
A bar chart with the number of genes and pseudogenes of FPRs in different vertebrates. The dataset was extracted from [137,139], and the raw data are available in Appendix A.

#### 3.4.2. Expression of FPRs

Like the majority of other CRs presented here, FPRs follow the one-receptor–one-neuron rule and are the only CR expressed in a given cell. Most FPRs are co-expressed with Gα_i2_, except FPR-rs1 in mice, which is co-expressed with Gα_o_ [133,134].

#### 3.4.3. Signaling Pathway of FPRs

FPRs share many features with VRs: they are expressed in the same type of neurons and co-expressed with the same G-proteins. Therefore, it is also thought that they share the same signaling pathway, beginning with the activation of PLC by the βγ subunit of the G-protein. PLC is responsible for the production of IP3 and DAG [117], which in turn activates the Na^+^/Ca^2+^ channel TRPC2, leading to the depolarization of the VSN [118,119]. This depolarization is amplified by the release of ER calcium ions [120] and by the activation of calcium-sensing chloride channels [121,122]. In addition, NHERF1 might be involved in the FPR pathway in the apical layer [125]. Negative feedback on TRPC2 with CaM activated by calcium ions results in a slow adaptation [126], and the inactivation of sodium ions results in a short-term adaptation [127].

There is also a mechanism of desensitization specific to FPRs caused by the GRK2-mediated phosphorylation of two clusters of residues, Ser328/Thr329 and Thr331/Ser332 for FPR1, leading to its internalization, but this has not been proven in VSNs [140].

#### 3.4.4. Structure and Binding Site of FPRs

Like the other receptors and GPCRs, FPRs have seven TM helices with the N-terminal part outside of the cell. The length of an FPR is approximately 350 amino acids, and like the other class A GPCRs, the binding site is localized inside the TM domain. More precisely, the D^3.33^ and the motif R^5.38^xxxR^5.42^ are crucial in the recognition of the formyl group of the ligand (Figure 16). FPRs show the same active state conformation as ORs, with the break of the ionic lock between R^3.50^ and D/E^6.30^ leading to the displacement of TM6. A new interaction then appears between R^3.50^ and Y^5.58^, which stabilizes this active conformation. There is also the “toggle switch”, which is a rotamer conformational change in W^6.48^; the residues I/V^3.40^, P^5.50^, and F^6.44^ also show conformational changes (Figure 17A) [141,142,143].

VNO-specific FPR-rs3 proteins have ten highly conserved residues in their sequences that are always absent in immune FPRs: H106, M121, E181, A247, I260, P272, S287, L300, E303, and K330 (Figure 16 and Figure 17B). Some of them are involved in the binding site, notably H106, and are also found in the other VNO-specific FPRs [138].

### 3.5. Taste Receptors (TRs)

#### 3.5.1. Genes and Evolution of TRs

TR genes were first identified in rat and mouse lingual epithelium through in situ hybridization in 1999 [144]. Indeed, two receptors of the GPCR family were identified and appeared to be related to the V2R family, with a long N-terminal extracellular domain characteristic of class C GPCRs. The two receptors were thought to belong to two different families because of their different localizations: TR1 in fungiform taste buds and TR2 in circumvallate taste buds. They were later identified as two members of the T1R family and named T1R1 and T1R2, respectively. A second study identified the second family of TRs, the T2R family, which is more related to V1Rs and opsins or class A GPCRs [23,145,146]. In fact, they are classified as an entire family, the class T2 GPCR family, because of important differences in structure and ligands between T2R and class A GPCRs [21,146]. Many members were first identified in the human genome, including a large proportion of pseudogenes, leading to an initial hypothesis of about 80 to 120 T2R genes, including a third of pseudogenes. The G-protein heterotrimer identified as being coupled with TRs is composed of α-gustducin, Gb1, and Gϒ13 [147].

TRs are hypothesized to have appeared with the emergence of the jaw apparatus, related to the diversification of feeding strategies [35]. T1R genes show a very stable evolution across vertebrates, especially in tetrapods. There are three T1R genes: T1R1A and T1R2, the umami and sweet subunits, respectively, and T1R3, the coreceptor (Figure 18). There are some exceptions, such as Actinopterygii, which shows another clade, T1R1B, or the ray-finned fishes, which have more T1Rs (up to 18) than other clades. T2R genes are more diversified to perceive bitter taste, with a significant expansion in amphibians for clades T2RE3 and T2RE5 and, to a lesser extent, in mammals for clades T2RE12 and T2RE13. The average number of T2R genes in mammals is 21.1, with a range from 0 to 54. Humans have 28 T2R genes and 10 pseudogenes contrasting with less than 10 T2R genes in birds and reptiles (Figure 19). Although diet seems to affect the number of T2R genes, with omnivores having the highest copy number, a clear conclusion cannot be drawn because this observation does not apply to birds, crocodiles, or ray-finned fishes [35].

T1R genes are located on chromosome 1 in humans and chromosome 4 in mice in a region of conserved synteny, and they have six exons encoding a protein of around 850 amino acids. T2R genes are located in clusters on chromosomes 5, 7, and 12 in humans and 2, 6, and 15 in mice, with only one exon encoding a protein of between 300 and 330 amino acids [148].

#### 3.5.2. Expression of TRs

The taste receptors are localized in taste buds, mainly in taste papillae, which are composed of type I, II, and III cells and basal cells. T1Rs and T2Rs are expressed in type II bud cells [148]. The expression of T1Rs and T2Rs is different. Indeed, T1R3 is co-expressed with another T1R, forming one heterodimer, T1R1+3 or T1R2+3, in the three types of papillae: circumvallate, foliate, and fungiform. Both dimers are also found in the palatal papillae. The T1R2+3 dimer is the sweet receptor capable of detecting a wide range of natural sugars or synthetic sweeteners, allowing the identification of energy-rich nutrients (Figure 18). The T1R1+3 dimer is the umami receptor, which detects L-amino acids [149,150,151]. On the other hand, T2Rs are expressed mainly in circumvallate and foliate buds but also in fungiform buds in a lesser extent. Each cell expresses a wide range of T2Rs. This type of expression allows mammals to detect many bitter substances, related to noxious and/or poisonous chemicals, without being able to distinguish between them [145,151,152].

#### 3.5.3. Signaling Pathway of TRs

Different pathways were initially proposed for T1Rs and T2Rs, for example, with cAMP [153]. However, these pathways were then thought to be more of a way of modulating the signal. Indeed, one pathway has been found to be predominant in the activation of TRs. Thus, the activation of TRs leads to the stimulation of the linked G-protein, activating phospholipase C β2 (PLCβ2), which hydrolyzes phosphatidylinositol 4,5-bisphosphate (PIP2) into DAG and IP3 [154]. IP3 is responsible for the release of Ca^2+^ from the ER, which directly or indirectly leads to the gating of transient receptor potential cation channel subfamily M members 4 and 5 (TRPM4/5). TRPM5 is responsible for taste neuron depolarization [155,156]. Nevertheless, the TRPM5 pathway is not the only major pathway for bitter and umami perception, as TRPM5 KO mice were still able to sense some bitter or umami compounds at the same level as the wild-type variant [157].

Interestingly, TRPM5 is temperature-sensitive, which may explain the enhanced sweetness perception with heating and the phenomenon of thermal taste in the absence of tastants [158].

#### 3.5.4. Structure and Binding Site of TRs

As for the VRs, the two TR families are similar to different CRs. In fact, T1Rs are similar to V2Rs, with around 850 residues, and T2Rs are similar to ORs and V1Rs, with approximately 310 amino acids. Consequently, T1R dimers recognize ligands through the VFT domain (Figure 20), which is part of the extracellular N-terminal region with the CRD, as in V2Rs [148]. T2Rs belong to the class T2 GPCR family, which is less studied than the other families, so information on their structures comes from recent studies on the T2R structure [159,160]. Indeed, these studies identified the two main residues involved in the binding site: W^3.32^ involved in π-π interaction and E^7.39^ involved in a salt bridge with the ligand. These two different types of interaction allow for the binding of a wide range of compounds. In the inactive state, ECL2 forms a helix that occupies the binding pocket. Activation of the T2Rs induces a conformational change in the side chain of Y^6.48^, but without the TM6 displacement observed in class A GPCRs. This change leads to a hydrophobic interaction between F^5.41^, Y^6.48^, and Y^7.45^ that stabilizes the active state. In addition, T2Rs also possess an allosteric binding site that is connected to the orthosteric site, and the bound tastant forms an interaction with the Gα protein (Figure 21).

### 3.6. Summary of CR Features

A table summarizing the main characteristics of each CR is shown in Table 1. Even though they are all GPCRs, CRs of different GPCR classes show important differences compared to CRs of the same GPCR family. The expression of all CRs involved in environmental sensing is clustered in different organs of the animal head.

## 4. In Silico Study of Structures

### 4.1. Modeling Theories

Proteins are sequences of amino acids that are linked together in a linear sequence by amide bonds, called a backbone, forming polypeptide chains. The amino acids can also interact via hydrogen bonds between the atoms of the backbone or the side chains. These interactions are responsible for the secondary structure (helices and sheets) and the global fold of the protein, or the tertiary structure. Some proteins work in associations of several chains or subunits held together by hydrogen bonds, hydrophobic interactions, or disulfide bonds. The protein needs the appropriate tertiary or quaternary structure to be functional [161]. Major advances in genome sequencing are increasing the number of available protein sequences faster than experimental methods can resolve the corresponding structures. Moreover, some interactions are not conserved with experimental methods, especially between the subunits of quaternary structures. To face this lack of structures compared to sequences, much effort has been dedicated to the development of computational structural models based on the knowledge of structures already obtained [162]. We present detailed information on the technical parts in Appendix A.

#### 4.1.1. Homology Modeling

Since 1967, the idea of homology or comparative modeling has been to use known structures as a template to create a model of a homologous protein [163]. This concept is based on three observations about conservation of protein structure, such as (i) The 3D structure of a protein is fully determined by its amino acid sequence; thus, only the sequence is necessary to predict the structure [164]. (ii) A small change will affect the structure on a small scale: proteins with over 50% identity show internal shifts of less than 1.5 Å deviation and conserve 85% of backbone angles [165]. (iii) Structure and function are more conserved than sequences: homologs tend to retain structural features better than engineered proteins with an equivalent sequence identity [166].

The standard protocol for homology modeling is composed of different steps, each of which can be executed by different algorithms (Figure 22A) [167,168,169,170,171]:Search for homologous structure by identifying proteins with high identity, common evolutionary origin, a high-quality resolved structure, and biological relevance (ligands, solvent, pH, and conformational state). Sequence identity thresholds depend on protein length (Figure 22B). Under this threshold, many random alignments are found, known as the twilight zone [172].The sequence alignment of target and template sequences; manual correction may be necessary, according to the bibliography.The generation of the backbone based on the coordinates of templates.Loop modeling for the gap regions or for poorly conserved regions, using either database-driven loop libraries or ab initio energy-based methods.Side-chain insertion based on rotamer libraries built from high-resolution structures and steric or energy constraints [173].The optimization of the model via energy minimization and/or dynamic simulation to resolve atomic clashes, abnormal bond lengths or angles, incorrect positions of loops, etc. The goal is to find a geometry with the minimum potential energy that corresponds to a state of equilibrium.Validation of the model on physics, knowledge, machine learning, or experiments scores, if the model fails, the alignment or template must be revised.

#### 4.1.2. Fold Recognition or Threading

Another template-based modeling method that emerged in the 1980s and 1990s is threading [174,175,176,177], which allows for the use of structural templates with low sequence identity [178,179,180,181]. Indeed, proteins can have a similar structure due to a common origin or evolutionary convergence. This strategy is efficient, as many proteins adopt a limited number of folds, fewer than initially thought [182], and a protein’s entire structure can often tolerate many substitutions if the essential residues are fixed [183].

As in homology modeling, threading aligns targets to templates; however, it incorporates more information than the amino acid sequence in an alignment matrix. Different types of algorithms are used to find templates in a fold database:Profile–profile alignment methods use homologs to identify the most important part of the sequence and to estimate the probability of a residue being at that position. This information is implemented in a sequence-position-specific scoring matrix (PSSM), called the sequence profile [184,185].A structural profile includes information on the environment of each amino acid in a score, including the secondary structure, the fraction exposed to polar atoms, and the fraction buried in the protein. Thus, environment-specific substitution tables have emerged to improve secondary structure predictions [177,181,186,187].Hidden Markov models (HMMs) are algorithms that create a probabilistic model, similar to a profile, that incorporates evolutionary events [188,189]. Iterations enhance the model’s ability to find distant homologs of a target and predict secondary structure, that also helps to improve the model’s ability to find real homologs because of the high conservation of the structure.

Machine learning approaches have been used to improve predictions by combining multiple sequence alignment (MSA), profile–profile alignment, structural profiles, and predicted secondary structures into a machine learning algorithm, such as a neural network or support vector machine [190,191]. These methods enable the selection of only the best templates and/or models obtained in the previous steps.

#### 4.1.3. Ab Initio or Template-Free

For 50 years, researchers have tried to solve the problem of protein folding with only sequence information. Indeed, it is assumed that the structural information is fully encoded in the sequence because it depends on atomic interactions [192,193].

Pure ab initio prediction methods have been developed following two different ideas. First, the objective is to fold the protein in steps, beginning with an extended or misfolded conformation without knowledge about related structures. Folding is performed iteratively following the thermodynamic hypothesis (using a free energy function and a score), which states that proteins fold in the global energy minimum [193,194,195,196,197]. To recreate the folding of a protein in a solvent over time, a method of simulating the molecules, called a molecular dynamics simulation (MDS), can be used. MDSs are based on the solving of equations of motion and a force field describing atomic interactions [198,199,200,201,202]. Second, the objective is to find the most probable conformation of the protein based on the physico-chemical properties of amino acids by studying the probability of each residue being in the secondary structure. This probability is calculated based on the neighbors and interactions, according to a statistical analysis of known structures [203,204,205,206]. Finally, AI-based methods have been developed to predict the secondary structure using neural networks or nearest-neighbor algorithms [207,208,209,210].

A new method, also considered ab initio, appeared later, but required more information than the amino acid sequence. Indeed, the fragment-based method assembles very short structural fragments unrelated to the target into various conformations [211,212]. Then, the different conformations are improved and sorted using an energy function. This method works as it is suggested that most protein folds exist in the PDB [213]. It quickly became one of the most successful ab initio strategies [214].

An original way to explore protein folding was created to allow people to participate, even without scientific knowledge, through the game Foldit [215].

#### 4.1.4. Deep Learning Methods

The evolution of machine learning has led to deep learning algorithms that can learn hierarchical data representation thanks to multiple processing layers, significantly improving existing methods for solving the folding problem. For example, it improves the fragment-based method by constructing better-quality fragment libraries specific to a protein [216,217]. Another key tool enhanced is the contact map, a matrix that represents pairs of amino acids distant in sequence but close in space, useful in the construction of 3D models [218,219,220,221]. Especially through artificial neural network-based methods [222] such as convolutional neural networks (CNNs) [223,224]. CNNs can detect patterns at different scales thanks to their convolutional and pooling layers [225]. More advanced distance-based methods now predict pairwise distances between residues without being constrained by a contact distance, allowing 3D model to be built via energy minimization according to the features of the distance map leading to a great improvement in protein prediction [226,227,228].

Early deep learning approaches mainly used supervised learning where the training set used is labeled, allowing the model to adjust its parameters [225]. However, unsupervised learning methods are becoming more efficient in some fields because they enable the algorithm to learn independently. With self-supervised learning, the algorithm can learn structural and functional information directly from the data, which makes it useful for predicting protein folding [229].

As transformers coupling a neural architecture for language (natural language processing or NLP) and attention had proved to be highly effective, bioinformaticians began to apply them to the folding problem [230]. Indeed, the sequence of 20 different amino acids in proteins can be interpreted as a language making NLP algorithms suitable for proteins [231,232]. The attention allows the model to connect distant residues in sequence that are spatially close, forming a contact map, and to identify binding site residues.

The development of deep learning methods has led to fully automated frameworks using the end-to-end technique in combination with several other powerful methods, and this has revolutionized the protein folding problem, resulting unprecedented accuracy. The principle of the end-to-end method is to use a single neural network that takes the raw data as input and can adapt to produce the expected outputs based on its training [233,234]. First, algorithms relied on MSA to obtain evolution information [235,236]; then, the development of single-sequence-based algorithms emerged. These require less computational time for the same quality of outputs [19,237,238]. However, even if these methods are highly effective, the training procedure requires extensive time and computational resources, which are not accessible to most research groups. Therefore, new methods with simplified architectures are emerging to reduce computational time without altering performance [239].

#### 4.1.5. Algorithm Comparison

Each method offers distinct advantages that must be considered when selecting an algorithm. Although recent advances in deep learning have yielded high quality models, confidence in the results can be improved with a comparative-based method if many homologous structures are available (Table 2). Ab initio methods, though parameter-dependent, are useful for studying protein folding. Also, the calculation time varies greatly between the algorithms with ab initio and MSA-based deep learning methods, being really time-consuming, and all deep learning methods require a long training [240].

### 4.2. Model Assessments

Since all modeling methods rely on estimations and probabilities, it is necessary to evaluate the model accuracy against experimental structures. For this purpose, algorithms can be tested blindly on sequences that will soon be released in the PDB. First, there is the Critical Assessment of protein Structure Prediction (CASP), a biennial initiative since 1994 that benchmarks protein structure prediction methods (https://predictioncenter.org/). Second, the Continuous Automated Model Evaluation (CAMEO) is a community-wide project based on the weekly release in the PDB; it also allows the performance of algorithms to be tested blindly, but more frequently in a fully automated way [241,242].

Two important metrics are used to evaluate the quality of a predicted model in comparison to the experimental structure: the global distance test total score (GDT_TS) and local distance difference test (lDDT). They both rely on an alignment of the structures and the use of different cut-offs: GDT_TS measures the percentage of Cα atoms pairs between the two structures that fall within predefined cut-offs and, lDDT compares all interatomic distances between the cut-off, evaluating the local environment [243,244]. Ramachandran plot can also be used to confirm the stereochemical quality of models, expecting more than 80% of residues in the core area of the plot as high-quality structure [245].

### 4.3. The Main Modeling Algorithms

Since the first model was released in 1967 [163], many algorithms have been developed and regularly improved (Figure 23). We present technical information on tools in Appendix A.

#### 4.3.1. Swiss-Model

The Swiss-Model was a pioneer in the field of automated modeling; indeed, it was the first fully automated protein homology modeling server [246]. The input is the sequence of the protein, which is used to search for templates in an internal template library [247]. The templates are selected and clustered based on their conformational states to identify the most suitable candidates. Finally, the coordinates of templates are used to generate the model, and the gaps are filled by loop modeling. A score based on the statistical potential of the mean force is provided with the models, which gives global and per-residue quality.

#### 4.3.2. Modeller

Modeller is an old comparative structure modeling program, having been created in 1993, but it is still updated [248]. Its algorithm is based on spatial restraints derived from known structures. Indeed, an alignment is made between the sequence to be modeled and known structures, then density probability functions are extracted from alignment information, and finally, the model is created by optimizing the probabilities. Since the seventh version of Modeller, a profile–profile alignment method has been implemented to improve the accuracy of predictions [249]. This software version requires the alignment of the sequence to be modeled with the template structures as an input. However, a web server is available where the Modeller algorithm only requires the amino acid sequence of the protein to be modeled [250].

#### 4.3.3. Rosetta

Released in the late 1990s, Rosetta was a pioneer in ab initio methods for structure prediction using fragments [212,251]. The results were rapidly promising for ab initio methods, with correct predictions in CASP experiments [214,252,253]. The combination of structural similarity search and Monte Carlo energy optimization allows for the creation of a de novo model for protein folds without a known structure [254,255,256]. Because of these works, David Baker was awarded the Nobel Prize in Chemistry in 2024 for computational protein design. New algorithms have been developed to extend the functionalities, such as comparative modeling [257,258], provision of more tools with ROSETTA3 [259], or deep learning methods. These include TrRosetta, a distance-based method using a convolution network to predict inter-residue orientations [260,261] and, RoseTTAFold with a three-track neural network for sequence, distance-map, and coordinate levels combining attention mechanisms and end-to-end training [235].

#### 4.3.4. Raptor

Raptor (Rapid Protein Threading by Operation Research) was a threading algorithm working with pairwise interactions using an integer/linear programming approach [262]. It was followed by RaptorX, a threading algorithm based on statistical learning with multiple templates, adjusting the reliance on profile or structural data based on input quality (available information and sequence profile) [263]. Then, deep learning methods led to RaptorX-Property, a web server that predicts structural properties without relying on templates, using deep convolutional neural networks and conditional neural fields [264]. This tool is part of a deep learning threading method with other algorithms able to improve the alignment and predict the interatom distance distribution for model creation [265]. The last tool developed is a single-sequence MSA-free structure prediction method, RaptorX-Single, based on the combination of three protein language models [266].

#### 4.3.5. TASSER

TASSER (Threading ASSembly Refinement) was a very promising threading-assembly method for ab initio prediction [267]. It used a hierarchical approach for template detection, the prediction of tertiary restraints, and a tertiary structure assembly by using Monte Carlo optimization of potentials. In 2007, TASSER was improved in I-TASSER, an iterative version of the method that improved the accuracy of prediction while remaining faster than ab initio methods [268]. Other improvements will follow [269] until deep learning-based protocols: C-I-TASSER which couples contact maps and the I-TASSER assembly protocol being useful for more proteins without homologs [270], and I-TASSER-MTD for multi-domain protein structures [271]. The various algorithms of TASSER have taken part multiple times in CASP experiments, consistently achieving top-ranked results. They are also freely available on the website of the Zhang lab [272].

#### 4.3.6. QUARK

The Zhang lab also proposes an ab initio method called QUARK. The algorithm combines fragment assembly (1 to 20 residues) and replica-exchange Monte Carlo simulation guided by an atomic-level knowledge-based force field to create a model with only the sequence. It was the best free-modeling algorithm in CASP9 and CASP10 [273]. Since then, IA has been implemented in this algorithm to improve models. Firstly, deep learning-based contact-map predictions were coupled to QUARK, leading to C-QUARK for contact-assisted QUARK [274]. Secondly, with the use of an MSA generated by deep learning, a distance map and dihedral-angle orientations could be created with a convolutional neural network. Finally, the model was still assembled with the QUARK method, but the method was called D-QUARK for distance-assisted QUARK [275]. These improved versions scored well in both CASP13 and CASP14.

#### 4.3.7. Medeller

Highlighting the importance of membrane proteins, a new homology modeling protocol was developed using a template structure aligned with the target sequence. Annotations on the alignment specify the position of the template in the membrane and the secondary structures. The modeling proceeds through different steps: the building of the core based on the template, loop prediction using FREAD [276] for the remaining gaps, and the option to fill the remaining large gaps [277].

#### 4.3.8. AlphaFold

One of the most important computational methods for predicting protein structure with high accuracy is AlphaFold2. Indeed, the AlphaFold2 team participated in CASP14 and ranked first, with a huge gap from second place, based on different metrics. For example, the metric used on the web page of CASP14 “https://predictioncenter.org/casp14/index.cgi (accessed on 24 February 2025)” is a sum of the GDT_TS without outliers (GDT_TS<−2). With this metric, AlphaFold2’s score is 244.0, whereas the second score is 92.1 for BAKER, the second team (Appendix A Appendix A) [278]. AlphaFold2 also provides models for 48 organism proteomes and the majority of Swiss-Prot referenced proteins “https://alphafold.ebi.ac.uk/download (accessed on 24 February 2025)”. Thanks to this breakthrough, Demis Hassabis and John Jumper were awarded the Chemistry Nobel Prize in 2024 for protein structure prediction.

AlphaFold2 is a deep learning method based on the attention mechanism and is composed of two transformers: Evoformer and the structure module [236]. Evoformer identifies the spatial interactions and the evolutionary relationship between residues. Then, the structure module creates a 3D model by moving the residues according to the information provided by the outputs of Evoformer. Three recycling cycles are performed using the outputs as new inputs to optimize the procedure. AlphaFold2 is the second version of the algorithm. Indeed, the first version, AlphaFold, was a distance-based method combined with optimization by a gradient descent algorithm [228]. AlphaFold had already outperformed the other methods in CASP13 [279]. The most recent version, AlphaFold3, is optimized for the prediction of complexes such as proteins, nucleic acids, ions, etc. [280].

#### 4.3.9. ColabFold

ColabFold is a freely accessible notebook on Google Colaboratory based on the Alphafold2 algorithm but using MMSEQ2 for the construction of the MSA instead of the initial methods of AlphaFold2. This difference can reduce the calculation time [281,282]. ColabFold performed well in the last CASP16 in protein domains on hard targets, ranking in 11th place.

#### 4.3.10. ESMFold

ESMFold is the first large-scale structural characterization of metagenomic proteins [19]. It also works without MSA, but instead of a complex transformer, it uses a simplified version of Evoformer. The efficiency is based on a large-scale language model with high complexity, with 15 billion parameters. Then, a structural transformer predicts the atomic coordinates. The simplified neural network architecture significantly improves runtime efficiency. Its fast calculation allowed the creation of an atlas of 772 million protein models “https://esmatlas.com/ (accessed on 5 March 2025)”.

## 5. CR Structural Studies

### 5.1. Experimental Structures

The first structure obtained for a vertebrate OR was OR51E2 [18]. This was obtained in 2023 using cryo-electron microscopy (cryo-EM) with a resolution of 3.10 Å (PDB ID: 8F76). This new structure shows great similarity to the OR models created before, validating the previous modeling protocols (Figure 24). It will be valuable for modeling new ORs based on homology and for validation. Then, three structures of a human consensus OR52 (OR52c) obtained using cryo-EM were also published in 2023 (PDB ID: 8HTI, 8J46, and 8W77) [283], followed by other human consensus ORs in 2024: consOR1 (PDB ID: 8UXY), consOR2 (PDB ID: 8UY0), consOR4 (PDB ID: 8UYQ), and consOR51 (PDB ID: 8UXV) [284].

Different structures are available for taste receptors. There are three human T2R46 structures, obtained in 2022 by cryo-EM in different states (PDB ID: 7XP4, 7XP5, and 7XP6) [153]. There are also many human T2R14 structures, obtained by cryo-EM in 2024 (PDB ID: 9IJA, 9IIW, 9IIX, 9IJ9 [275], 8VY7, 8VY9 [154], 8XQL, 8XQN, 8XQO, 8XQP, 8XQR, 8XQS, 8YKY, 8XQT [276], 8RQL [285]). In addition, there are different human T1R structures, some are incomplete with only the transmembrane part (T1R2: 9NOY and 9NOX, T1R2-T1R3: 9O38) or VFT part (T1R2-T1R3: 9NOU, 9NOV, 9NOW), and three are complete (9NOR, 9NOS, and 9NOT) [286]. Finally, there are five fish (*Oryzias latipes*) dimer T1R2a-T1R3 structures, obtained by X-ray diffraction in 2017 with different ligands (PDB ID: 5X2M, 5X2N, 5X2O, 5X2P, and 5X2Q) [277].

Many structures obtained using cryo-EM are available for human FPRs. Indeed, five FPR1 structures (PDB ID: 7T6T [278], 7VFX and 7EUO [136], 7WVU [137]) and nine FPR2 structures are accessible (PDB ID: 6LW5 [138], 6OMM [279], 7T6S, 7T6U, 7T6V [278], 7WVV, 7WVW, 7WVX, 7WVY [137], 8Y62, 8Y63 and 9JHJ [287]). However, no structure is referenced in the PDB for other species.

A lot of TAAR structures were released in 2023, all obtained using cryo-EM, but most of them were not involved in chemoreception (TAAR1). Indeed, three TAARs were studied: human and murine TAAR1, murine TAAR7f, and TAAR9. The PDB IDs are listed below:TAAR1: ◦Mouse: 8WCB, 8WCC, 8WC3, 8WC4, 8WC5, 8WC6, 8WC7, 8WC9 [288], 8JLJ, 8JLK [289] 8ZSV [290];◦Human: 8WCA, 8WC8 [288], 8JLN, 8JLO, 8JLP, 8JLQ, 8JLR, 8JSO [289], 8UHB [82], 8W87, 8W88, 8W89, 8W8A [291], 8ZSJ, 8ZSP, 8ZSS [290] 9JKQ [292];
Murine TAAR7: 8PM2 [79];Murine TAAR9: 8ITF, 8IWE, 8IWM, 8IW1, 8IW4, 8IW7, 8IW9 [293].

There is no structure elucidated for VRs—neither type 1 nor type 2.

In the past two years, numerous CR structures have been released, and more are scheduled for publication in 2025, marking a significant turning point in structural studies for membrane proteins. This progress is due to improvements in cryo-EM techniques and a better knowledge of the CRs.

### 5.2. Predicted Models

Efforts on modeling CR are principally on ORs and TRs to study the ligand interaction, their binding site, and their activation [286,287,288,289,290,291,292,293,294,295,296]. Indeed, ORs and TRs are key to smell and taste, respectively, which are widely studied for sensory research and applications in perfume or food industries. They are also involved in a lot of other biological functions in different organs, making them important targets for medical and physiological studies [35,297].

FPRs and TAARs are also interesting targets for computational studies due to their implication in biological processes. Indeed, FPRs are involved in inflammatory response and host defense in addition to their chemoattractant role; thus, modeling studies are conducted to study their structures [298,299,300]. TAARs are implicated in homeostasis and are important targets for psychiatric disorders, especially TAAR1 and TAAR5, leading to computational studies of the interactions and activation mechanism [69,76,301,302,303].

However, only a few studies have been performed with the aim of modeling VRs, probably because their role is more limited to chemical communication than other CRs, and because of their weak presence in humans [80,304,305].

Most of CRs modeling studies were performed using homology modeling. In the future, with the development of fast and accurate tools described above, it is reasonable to expect that methods will change in favor of new end-to-end algorithms. Many AlphaFold2 models are also already available in the UniProt database for all types of CRs, to different extents, and for different species.

### 5.3. Limitations

Despite all these improvements in experimental methods, the limitations presented earlier still exist for many CRs. For instance, only one real OR structure is available, and it was chosen because of its favorable characteristics, such as its expression outside OSN or its ability to bind water-soluble ligands [18]. Moreover, experimental structures also show issues like gaps in loop regions due to high flexibility or the lack of some intermittent conformational states [294].

Even if algorithms are in a state of constant improvement, the protein folding problem is not resolved as some limitations still exist. Indeed, the training of algorithms is limited by the lack of experimental structures and as knowledge of some types of folds and is biased by the presence of unnatural folds in the PDB because of the special conditions used in the experiments [295,296]. Moreover, the long-loop regions cause real difficulties when being modeled, mainly because of their flexibility [297]. There are also issues in the modeling of membrane positions because algorithms often create models without the information about the membrane position or a too simplified implicit membrane [298].

## 6. Usefulness of Structural Studies

The combination of structural studies and deorphanization work performed on CRs (all the studies aimed at finding ligands for CRs) provides extensive information on binding sites, protein–ligand interactions, and structure–activity relationships [299]. Thanks to this information, ligand-based and structure-based pharmacophore analyses are beginning to be conducted on CRs [300,301,302]. A ligand-based pharmacophore is a simplified model that groups the essential common chemical features that are inherent in a set of known ligands for the same macromolecule. This model is specifically designed to illustrate the essential aspects governing the interactions between these ligands and the target. A structure-based pharmacophore is a description of the binding site that includes both spatial and electronic features. This type of pharmacophore elucidates the specific molecular environment within the binding site, which is conducive to optimal ligand interaction [303,304]. These methods enhance the knowledge on CRs and their interactions with ligands faster than in vitro and in vivo studies. A better understanding of the interactions between CRs and ligands would help medical research. For example, many research studies focus on ORs because of their implications for many biological processes: glucose homeostasis, blood pressure regulation, the inhibition of tumor progression, etc. [37]. It would also help semiochemical research, which aims to improve interactions between living beings by using the natural language of chemical communication. For instance, semiochemical products can replace insecticides [305] or improve the well-being of domestic and farm animals [306,307,308]. In the coming years, artificial intelligence and data science approaches are expected to play an increasingly important role in accelerating model generation, ligand prediction, pharmacophore analysis, and the interpretation of complex structural datasets.

## Figures and Tables

**Figure 1 ijms-26-06605-f001:**
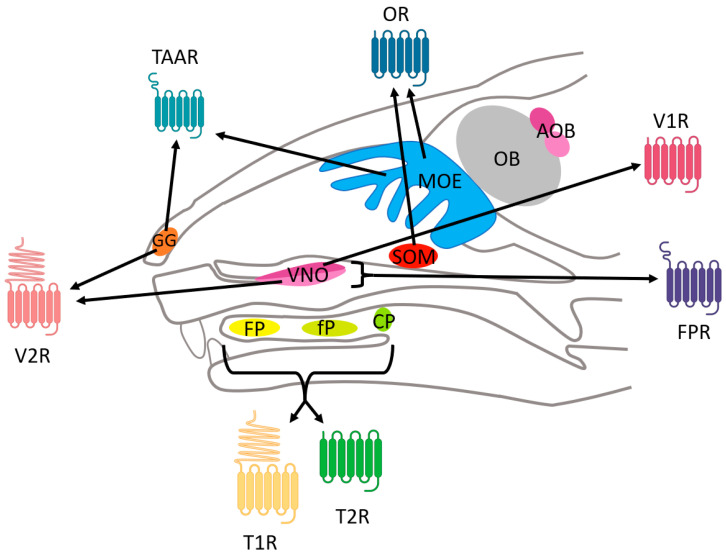
A schematic representation of mouse chemosensory systems. Accessory olfactory bulb (AOB), circumvallate papillae (CP), foliate papillae (fP), formyl peptide receptor (FPR), fungiform papillae (FP), Grueneberg ganglion (GG), main olfactory epithelium (MOE), olfactory bulb (OB), olfactory receptor (OR), septal organ of Masera (SOM), taste receptor type 1 (T1R), taste receptor type 2 (T2R), trace amine-associated receptor (TAAR), vomeronasal organ (VNO), vomeronasal type 1 receptor (V1R), and vomeronasal type 2 receptor (V2R).

**Figure 2 ijms-26-06605-f002:**
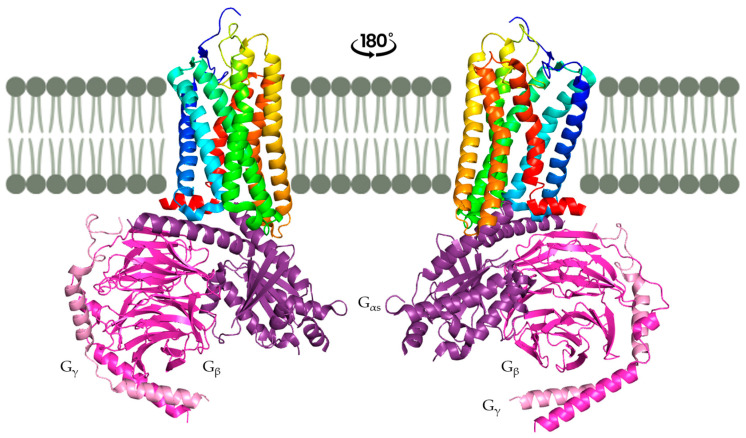
The structure of human OR51E2, a class A GPCR, coupled to the G-protein miniG_s399_ (PDB ID: 8F76); the structure was released in 2023. The helical bundle conformation specific to GPCRs is visible in the membrane region of the protein. OR: rainbow color; Gαs: purple; Gβ: pink; Gγ: light pink. Visualization with Maestro v14.2.121 (Schrodinger, LLC., Portland, OR, USA).

**Figure 3 ijms-26-06605-f003:**
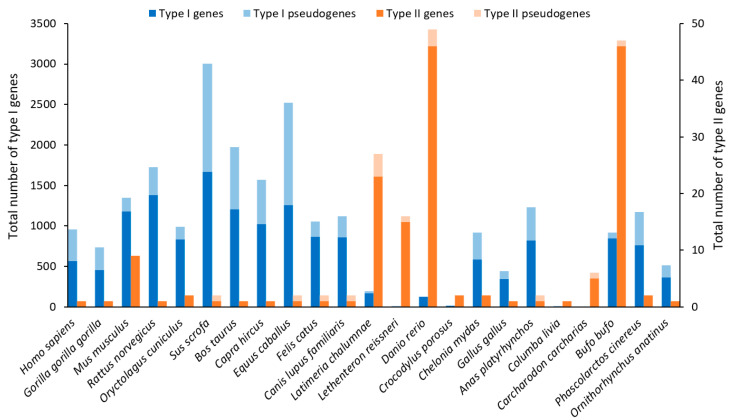
A bar chart with the number of genes and pseudogenes in OR classes in different vertebrates. The dataset was extracted from Policarpo et al. [35], and the raw data are available in the Appendix A.

**Figure 4 ijms-26-06605-f004:**
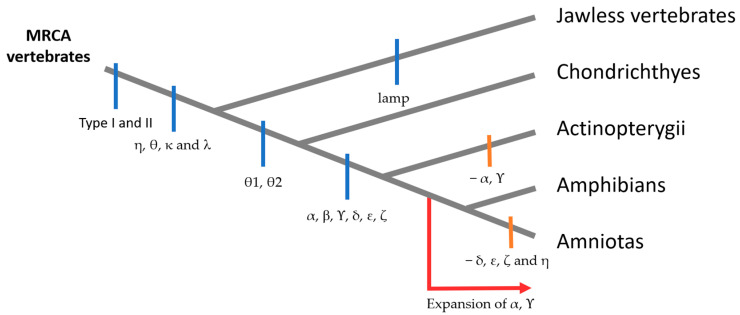
A schematic illustration of the evolution of OR gene families in vertebrates. In the tetrapod lineage, the number of group α and Υ genes has dramatically increased, probably due to the importance of olfactory information in terrestrial life. MRCA: most recent common ancestor. Adapted and modified from [33,41].

**Figure 5 ijms-26-06605-f005:**
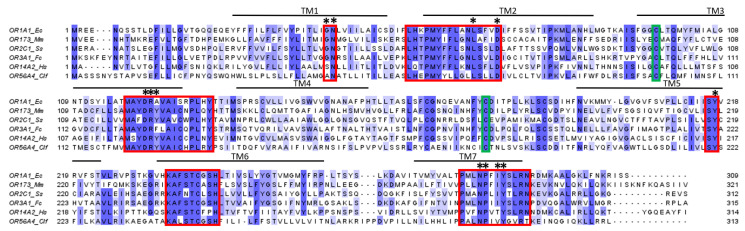
The alignment of different ORs from different species, highlighting conserved motifs. Residue conservation greater than 30% is marked in blue, and intensity is proportional to conservation. Red boxes highlight conserved motifs of ORs. The asterisks (*) indicate conserved residues in all class A GPCRs. Green boxes show conserved cysteines involved in the disulfide bridge. Ec: *Equus caballus*; Mm: *Mus musculus*; Ss: *Sus scrofa*; Fc: *Felis catus*; Hs: *Homo sapiens*; Clf: *Canis lupus familiaris*. Alignment was performed with MAFFT and viewed with Jalview v2.11.4.1.

**Figure 6 ijms-26-06605-f006:**
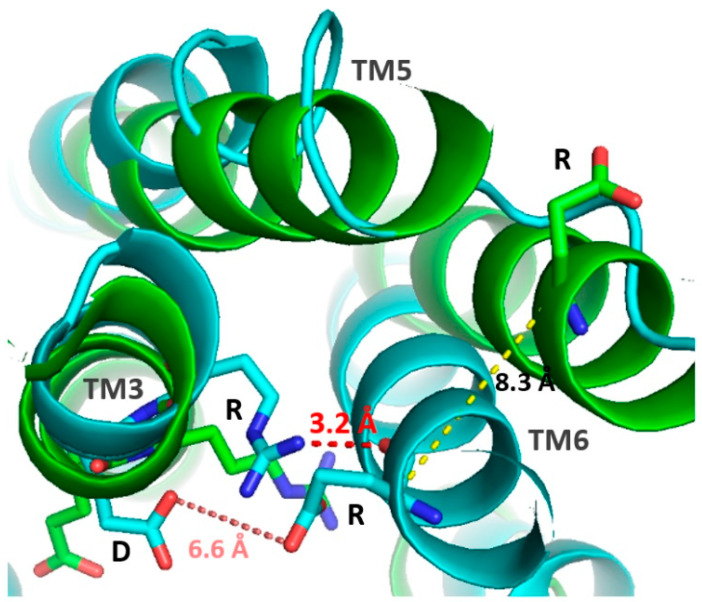
A comparison of ionic bridges (in red) and TM6 displacement (in yellow) of active (green, PDB ID: 2X72) and inactive (cyan, PDB ID: 1U19) bovine rhodopsin, a class A GPCR. The residues shown are those involved in the ionic bridge with carbons in the same color as the structure, the oxygen atoms are red, and the nitrogen atoms are blue. The alignment of structures and distances was calculated on Maestro v14.2.121 (Schrodinger, LLC., Portland, OR, USA) with default parameters.

**Figure 7 ijms-26-06605-f007:**
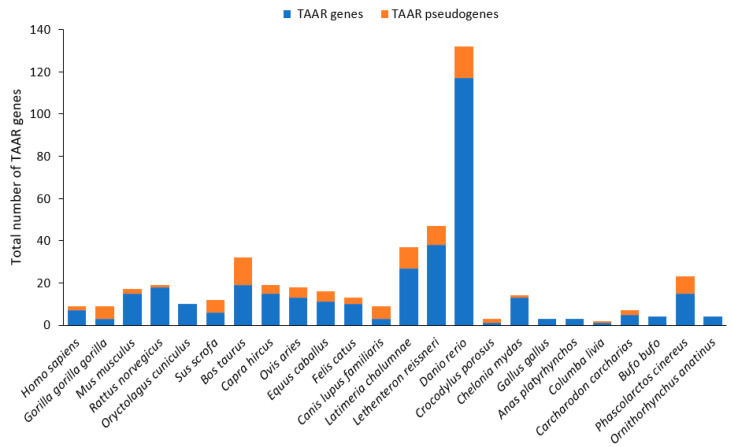
A bar chart with the number of genes and pseudogenes of TAARs in different vertebrates. The dataset was extracted from Policarpo et al. [35], and the raw data are available in Appendix A.

**Figure 8 ijms-26-06605-f008:**
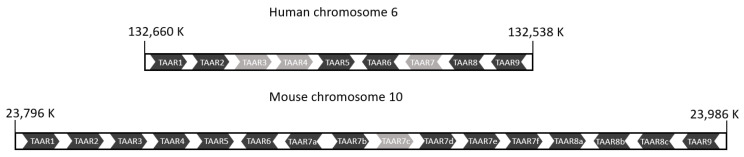
The orientation of human and mouse TAAR genes. Genes are represented by dark gray arrows, and pseudogenes by light gray arrows. The orientation of the arrow shows the reading direction. The width and distance are not to scale. Note that the human chromosome is displayed in reverse orientation.

**Figure 9 ijms-26-06605-f009:**
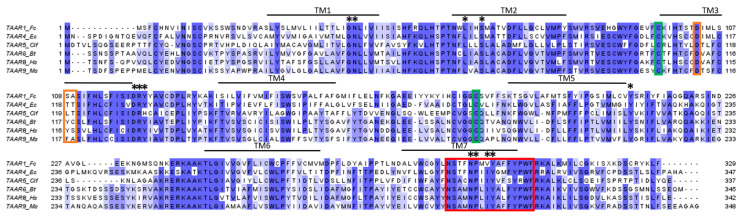
The alignment of different TAARs from different species, highlighting conserved motifs. A residue conservation greater than 30% is marked in blue, and intensity is proportional to conservation. The asterisks (*) indicate conserved residues in all class A GPCRs. Green boxes show conserved cysteines involved in the disulfide bridge. The red box highlights the conserved motif of TAARs. Orange boxes indicate residues specifically involved in the binding site of TAARs. Fc: *Felis catus*; Ec: *Equus caballus*; Clf: *Canis lupus familiaris*; Bt: *Bos taurus*; Hs: *Homo sapiens*; Mm: *Mus musculus*. Alignment was performed with MAFFT and viewed with Jalview v2.11.4.1.

**Figure 10 ijms-26-06605-f010:**
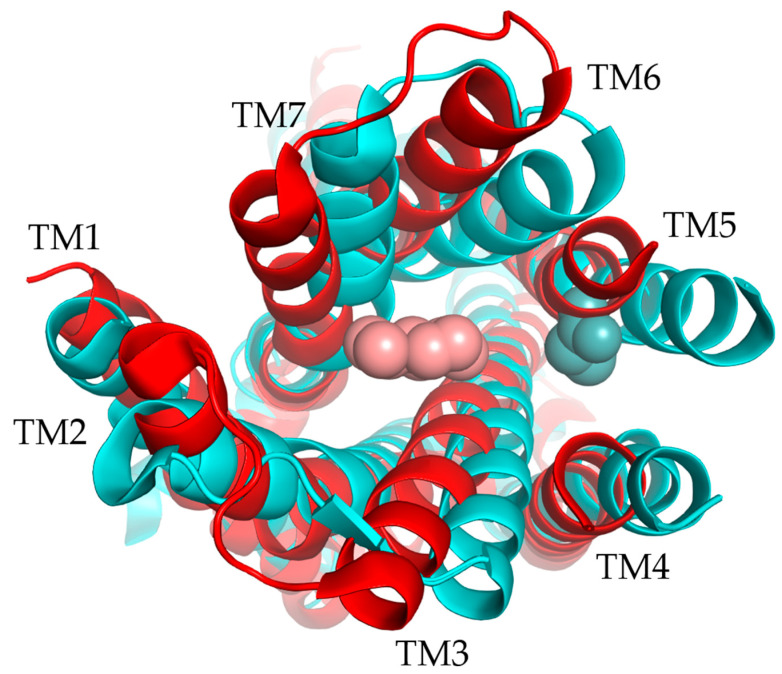
The alignment of murine TAAR7f in red (PDB ID: 8PM2) and human OR51E2 in blue (PDB ID: 8F76) with their ligands in spheres showing the different binding sites. ECL2 and N-terminal loops are hidden. The alignment of structures was calculated on Maestro v14.2.121 (Schrodinger, LLC., Portland, OR, USA) with default parameters.

**Figure 11 ijms-26-06605-f011:**
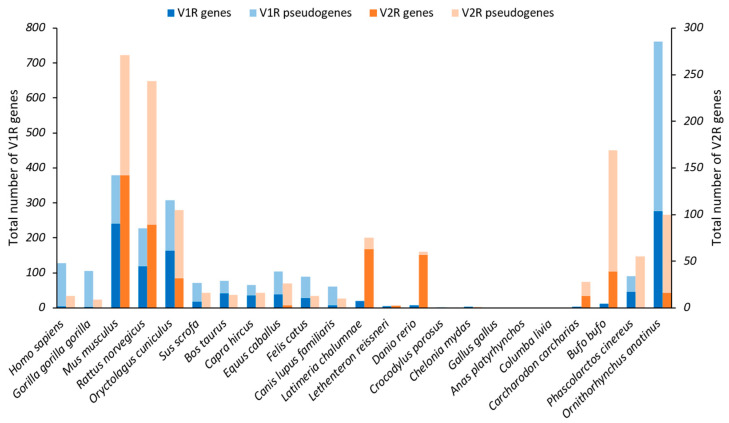
A bar chart with the number of genes and pseudogenes of VR families in different vertebrates. The dataset was extracted from Policarpo et al. [35], and the raw data are available in Appendix A.

**Figure 12 ijms-26-06605-f012:**
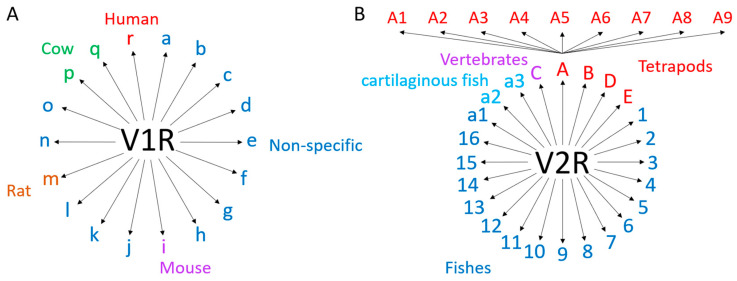
VR families and subfamilies of different species. (**A**) V1R classification with species-specific subfamilies. Some VRs are not yet well classified; thus, the species specificity could be updated in the future. (**B**) V2R classification in vertebrates.

**Figure 13 ijms-26-06605-f013:**
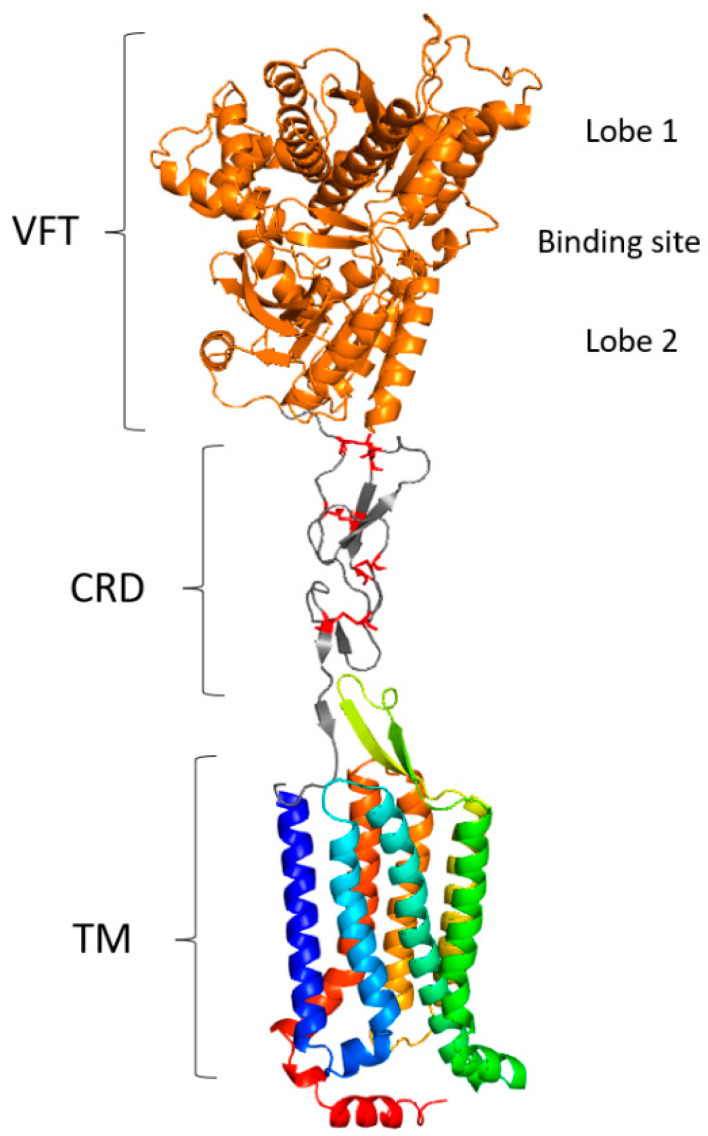
Mouse V2R116 Alphafold prediction (UniProt ID: E9Q6I0) highlighting the structure of the binding site with lobes. Residues shown in red are conserved cysteines. CRD: cysteine-rich domain; VFT: Venus flytrap domain; TM: transmembrane. Visualization with Maestro v14.2.121 (Schrodinger, LLC., Portland, OR, USA).

**Figure 14 ijms-26-06605-f014:**
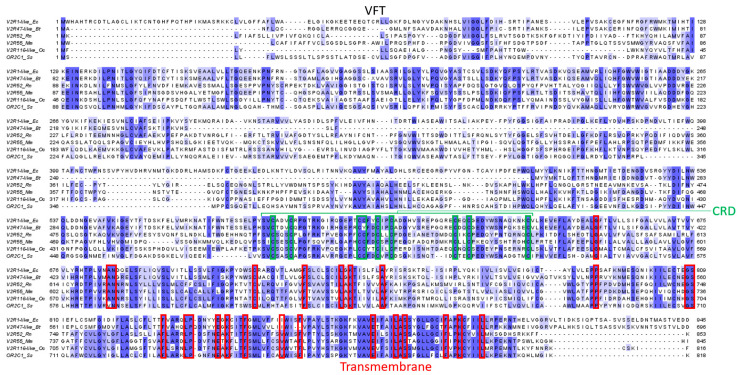
The alignment of different V2Rs from different species, highlighting conserved residues. Residue conservation greater than 30% is marked in blue, and intensity is proportional to conservation. Red boxes highlight conserved transmembrane residues of V2Rs. Green boxes show conserved cysteines involved in disulfide bridges in the CRD. Ec: *Equus caballus*; Bt: *Bos taurus*; Rn: *Rattus norvegicus*; Mm: *Mus musculus*; Oc: *Oryctolagus cuniculus*; Ss: *Salmo salar*. Alignment was performed with MAFFT and viewed with Jalview v2.11.4.1.

**Figure 16 ijms-26-06605-f016:**
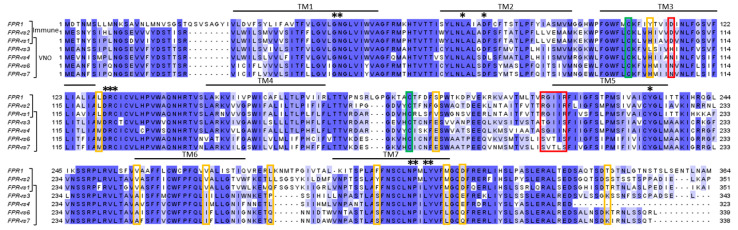
The alignment of murine FPRs highlighting conserved residues. Residue conservation greater than 30% is marked in blue, and intensity is proportional to conservation. The asterisks (*) indicate conserved residues in all class A GPCRs. Red boxes highlight residues involved in the recognition of the formyl group of the ligand. Green boxes show conserved cysteines involved in the disulfide bridge. Yellow boxes show residues conserved only in VNO-expressed FPRs. Alignment was performed with MAFFT and viewed with Jalview v2.11.4.1.

**Figure 17 ijms-26-06605-f017:**
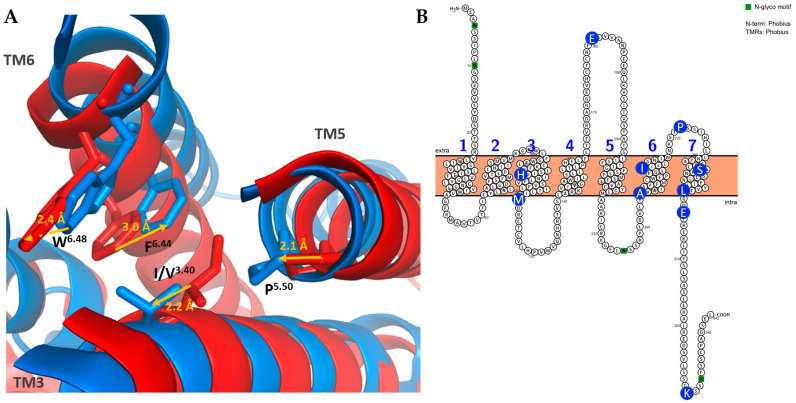
FPR analysis of structures and sequences. (**A**) Conformational changes in the residues W^6.48^, P^5.50^, I/V^3.40^, and F^6.44^ in human FPR2 in blue (PDB ID: 7WVW, active) and µ-opioid receptor in red (PDB ID: 4DKL, inactive), sharing similar conformations. The orange arrows show the displacement of an atom in each residue. Alignments of structures and distances were calculated on Maestro v14.2.121 (Schrodinger, LLC., Portland, OR, USA) with default parameters. (**B**) The FPR-rs3 sequence (NP_032066.2) with 7TM was estimated using Protter v1.0 and the 10 conserved residues specific to FPR-rs3 of different rodents.

**Figure 18 ijms-26-06605-f018:**
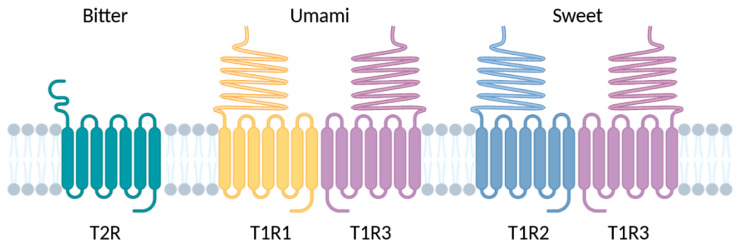
A summary of the different tastes, bitter, umami, and sweet, perceived with the T1Rs in dimers or T2Rs, all located in a membrane of taste bud cell.

**Figure 19 ijms-26-06605-f019:**
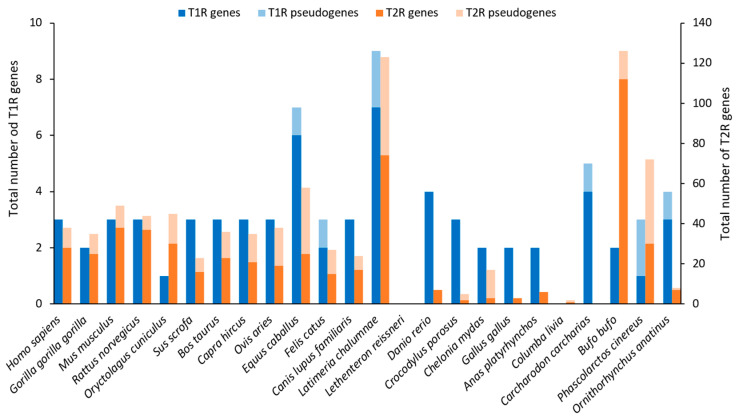
A bar chart with the number of genes and pseudogenes of TR families in different vertebrates. The dataset was extracted from Policarpo et al. [35], and the raw data are available in Appendix A.

**Figure 20 ijms-26-06605-f020:**
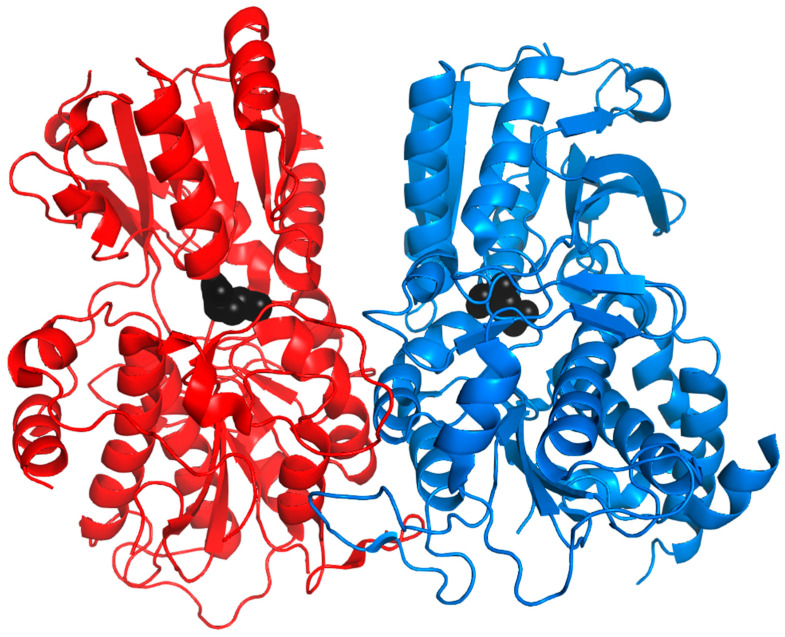
The VFT domain of the heterodimer T1R2a (red) + T1R3 (blue) in the medaka fish with glutamine (in black) in the binding site (PDB ID: 5X2M). Visualization with Maestro v14.2.121 (Schrodinger LLC., Portland, OR, USA).

**Figure 21 ijms-26-06605-f021:**
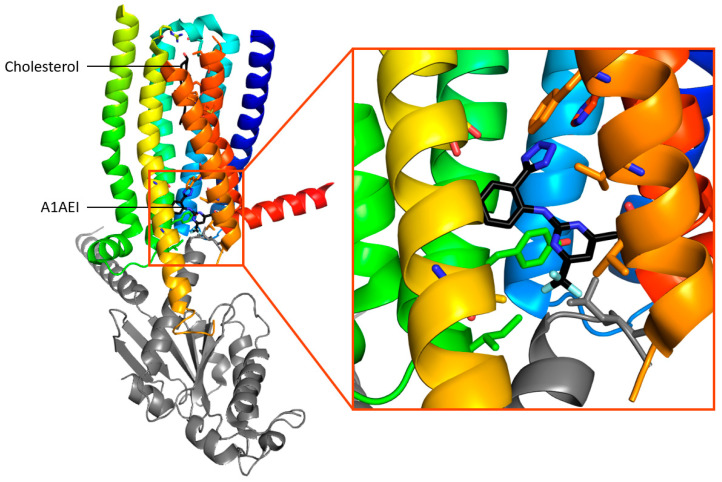
**The** Cryo-EM structure of human T2R14 (in rainbow color) coupled to the Gα_i_ protein (gray) with cholesterol in the orthosteric binding site and the tastant A1AEI in the allosteric binding site (PDB ID: 8VY7). The allosteric binding site is zoomed in. The residues that interact with ligands are shown. Visualization was performed with Maestro v14.2.121 (Schrodinger, LLC., Portland, OR, USA).

**Figure 22 ijms-26-06605-f022:**
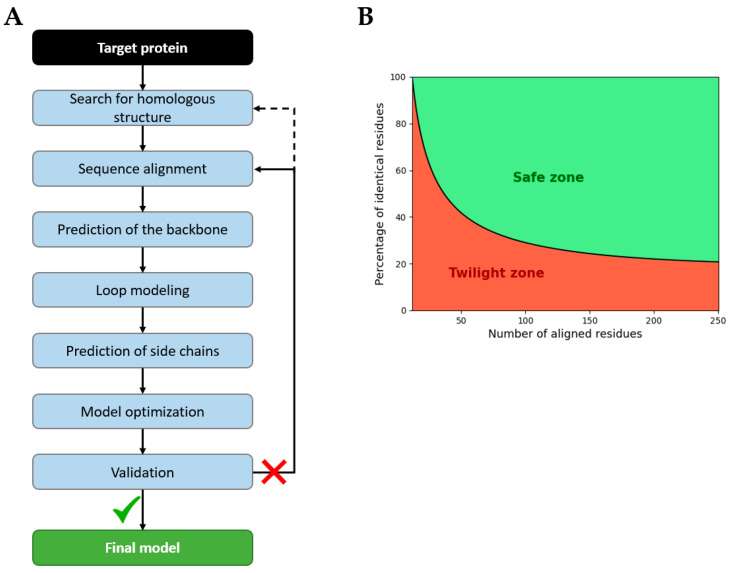
The homology modeling protocol. (**A**) The detailed step-by-step protocol, which can be repeated to correct the alignment. (**B**) A plot of the limit of the twilight zone of the identity percentage as a function of the length of the proteins. The safe zone is the identity percentage where the alignment is not considered random.

**Figure 23 ijms-26-06605-f023:**
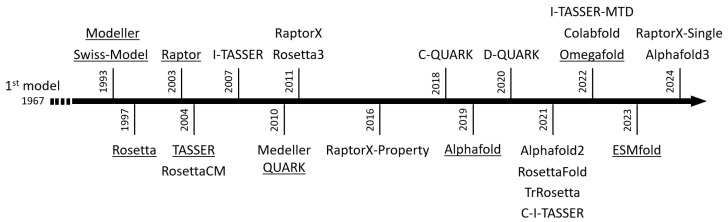
Timelines of creation of the main modeling algorithms (underlined) and their updates over time. Timeline not to scale for readability.

**Figure 24 ijms-26-06605-f024:**
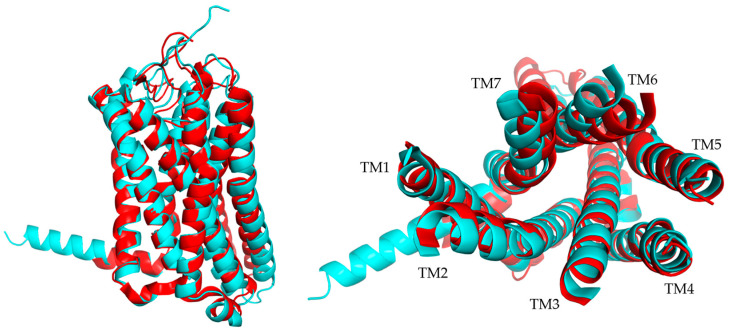
Comparison between human OR51E2 structure (PDB ID: 8F76), shown in red, and its AlphaFold prediction (AF ID: AF-Q9H255-F1), shown in blue. Extracellular loops are hidden from view on N-terminal side (right side). Alignment highlights high-quality prediction of AlphaFold, especially in TM region, compared to structure. Alignment of structures was calculated on Maestro v14.2.121 (Schrodinger, LLC., Portland, OR, USA) with default parameters.

**Table 1 ijms-26-06605-t001:** Comparison of sequence and structural properties of different CRs.

			VR			TR	
	OR	TAAR	V1R	V2R	FPR	T1R	T2R
GPCR Class	A	A	A	C	A	C	T
Length	~310	~350	~310	~850	~350	~850	~300–330
N-terminal	short	short	short	long	short	long	short
Exon	1	1–2	1	~6	~1–3	6	1
Expression involved in sensing	MOE/SOM	MOE/GG	VNO apical	VNO basal/GG	VNO	FP, fP, CP	FP, fP, CP
Binding site	TM3 TM5 TM6 ECL2	TM3 TM5–7 ECL2	TM3 TM5 TM6 ECL2	VFT	TM3 TM5	VFT	TM3 TM7

**Table 2 ijms-26-06605-t002:** Comparative analysis of various modeling algorithms.

Type of Algorithm	Principle	Advantages	Disadvantages
Homology	Modeling by comparison with structures of homologous proteins	Most accurate and reliable results when homologs are available	Limited to available structures Mistakes can arise from errors in alignment
	Modeller, Swiss-model, RosettaCM, Medeller		
Threading	Use both structures of similar proteins and sequences with structural information	Less limitation in sequence similarity than homology	Limited to available folds
	Raptor, Tasser, I-Tasser		
Ab initio	Create models based on biophysical principles (total energy, interactions, angles, …)	Can create new types of structures and folds Give information on the folding process	Requires a lot of computational resources
	Rosetta, QUARK		
Deep learning	Use known structures to learn how to fold proteins	Can create new types of structures and folds High accuracy Fast	Lacks transparency Requires extensive computational resources to train
	RaptorX, RaptorX-Single, AlphaFold2, AlphaFold3, TrRosetta, RosettaFold, C-I-Tasser, I-TASSER-MTD, Omegafold, ESMfold, C-QUARK, D-QUARK		

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
