# Peer review of "Chemosensory Receptors in Vertebrates: Structure and Computational Modeling Insights"

_ijms, 2025, doi:10.3390/ijms26146605_

Round 1

Reviewer 1 Report

Comments and Suggestions for Authors

Lamy et al aims to provide a comprehensive review of chemosensory receptor genes and protein in mammals. While this effort is commendable as an all-encompassing review of CRs is not available or is dated.  There are a few minor errors and missing information that I wish to point out. While discussing methods for determining CR structure, it is helpful to mention cryo-electron microscopy in the introduction section, as this recent method has generated the vast majority of structures. With respect to the olfactory system, they have left out the septal organ altogether. In section 3.5.2 and table 1, the expression data for T1rs and T2rs is inaccurate. They are expressed in all taste fields, with way less dramatic differences between papillae than the authors assert. Adding a section on the recently published structure of the human sweet taste receptor will be very helpful for readers. Finally, much of the information in section 4 is superfluous, and it is suggested that this be shortened to maintain focus on CR structure and function. A few other points are mentioned below. 

Line 64- G protein classification is in fact based on the g alpha subunit 

Line 78: T1rs belong to Class C. 

Section 4.1.1 needs to be rewritten for clarity 

Author Response

First of all, we would like to thank the reviewers for their valuable comments and corrections on our manuscript, which have greatly improved its quality. We have carefully revised/modified the manuscript to address all the queries raised by the reviewers. These modifications will not influence the framework of the manuscript. The changes were displayed in the revised manuscript in revision mode and highlighted in the PDF file. We have also responded point-by-point to each of your comments and suggestions. The manuscript was edited for the language, spelling, and grammatical errors by MDPI service. We appreciate your warm work earnestly and hope that the correction will meet with approval.

Lamy et al aims to provide a comprehensive review of chemosensory receptor genes and protein in mammals. While this effort is commendable as an all-encompassing review of CRs is not available or is dated.  There are a few minor errors and missing information that I wish to point out.

Comment 1: While discussing methods for determining CR structure, it is helpful to mention cryo-electron microscopy in the introduction section, as this recent method has generated the vast majority of structures.

Response 1: Thank you for your valuable suggestions. We agreed and added a few sentences (Line 59-63) about Cryo-EM and its importance in membrane protein structure in the introduction, after the mention of X-ray and NMR.

Comment 2: With respect to the olfactory system, they have left out the septal organ altogether.

Response 2: Thank you for pointing this out. We have corrected the text in the introduction and incorporated the septal organ in Figure 1 and Table 1, which was inadvertently excluded from the previous version of the manuscript. Lines 33-34 / 38 / 44 / 52

Comment 3: In section 3.5.2 and table 1, the expression data for T1rs and T2rs is inaccurate. They are expressed in all taste fields, with way less dramatic differences between papillae than the authors assert.

Response 3: Thank you for the interesting comment. We have decided to adopt a less restrictive approach regarding the expression of taste receptors (TRs) in Section 3.5.2, Figure 1 and Table 1. We now present the expression profiles of all TRs across all taste fields, highlighting that T2Rs are more prominently expressed in circumvallate and foliate taste buds. Lines 668-670 / 673-674

Comment 4: Adding a section on the recently published structure of the human sweet taste receptor will be very helpful for readers.

Response 4: As per your suggestion. The list of structures with PDB ID has been updated with the recent structures in Section 5.1, including the nine human sweet taste receptors structures from 2025, but also one T2R and three FPRs. Lines 1028-1031 / 1043

Comment 5: Finally, much of the information in section 4 is superfluous, and it is suggested that this be shortened to maintain focus on CR structure and function.

Response 5: Thank you for your comment. We have modified and shortened Section 4 to maintain the focus on CRs according to your suggestions. We have kept the most important information about the modeling algorithm mechanisms in the main text and transferred the technical information of the tools to Supplementary Information File 1 for all the methods in Section 4.

We have also added a few lines for algorithm comparison and evaluation in response to a suggestion from reviewer 2. Lines 869-875 / 887-894

Comment 6: A few other points are mentioned below. Line 64- G protein classification is in fact based on the g alpha subunit 

Response 6: Thank you for the indications, the sentence has been corrected to highlight the role of G-alpha subunit in the classification. Lines 79

Comment 7: Line 78: T1rs belong to Class C. 

Response 7: Thank you for the correction. We have incorporated the correction in Line 88.

Comment 8: Section 4.1.1 needs to be rewritten for clarity 

Response 8: Thank you for your suggestion. We have rewritten section 4.1.1 to make it easier to understand.

Reviewer 2 Report

Comments and Suggestions for Authors

This manuscript reviews vertebrate chemosensory receptors and computer modeling. However, it has some issues that need fixing before it can be published. First, the title and abstract are too broad. They don't clearly show the focus on GPCRs or the specific use of computer modeling. The introduction doesn't explain well why in silico methods are used or the scientific challenges. The 'results' section describes receptor analysis but doesn't critically analyze it. It doesn't discuss unresolved issues, gaps in knowledge, or model limitations. Figures are not well connected to the main points and are often mentioned without explanation.

The main problem is in the methods section. It does not clearly explain the modeling tools used. There is no comparison of methods like homology and AI-based predictions. It also lacks evaluation metrics for model quality. If the authors used structures they visualized or aligned, they need to explain how and with what software. The discussion and conclusion just repeat earlier points and do not offer new insights into the challenges of predicting structures for different CR families. They also do not suggest a useful research plan. The paper does not address the limitations of in silico methods, such as errors in loop regions, membrane placement, or ligand interactions. This makes it incomplete for a high-level journal. In short, while the review has a lot of content, it needs major changes to meet academic standards. It needs more technical detail, clear methods, figures, and a more critical and insightful discussion.

Author Response

First of all, we would like to thank the reviewers for their valuable comments and corrections on our manuscript, which have greatly improved its quality. We have carefully revised/modified the manuscript to address all the queries raised by the reviewers. These modifications will not influence the framework of the manuscript. The changes were displayed in the revised manuscript in revision mode and highlighted in the PDF file. We have also responded point-by-point to each of your comments and suggestions. The manuscript was edited for the language, spelling, and grammatical errors by MDPI service. We appreciate your warm work earnestly and hope that the correction will meet with approval.

This manuscript reviews vertebrate chemosensory receptors and computer modeling. However, it has some issues that need fixing before it can be published.

Comment 1: First, the title and abstract are too broad. They don't clearly show the focus on GPCRs or the specific use of computer modeling.

Response 1: Thank you for the comment. We agree on the lack of clarity about the subject in the abstract of our review. We have therefore modified the title of this review as follows: “Chemosensory receptors in vertebrates: Structure and Computational Modeling Insights” to make it clearer and more precise. We have also added a mention of GPCRs in the abstract to highlight their presence in the review, as well as a sentence on computer modelling. However, we wanted to keep the focus specifically on chemosensory receptors rather than on all GPCRs in the title. Lines 11-12 / 18-19

Comment 2: The introduction doesn't explain well why in silico methods are used or the scientific challenges.

Response 2: Thank you for the correction. Sentences have been added to emphasis the challenges of experimental methods, including cryo-EM, leading to the use of in silico modeling that are less limited and faster to produce. Lines 63-67

Comment 3: The 'results' section describes receptor analysis but doesn't critically analyze it. It doesn't discuss unresolved issues, gaps in knowledge, or model limitations.

Response 3: Thank you for the suggestions, a new paragraph have been added “5.3 Limitations” to highlight the issues and limits in CR structures and models. Lines 1082-1096

Comment 4: Figures are not well connected to the main points and are often mentioned without explanation.

Response 4: Thank you for pointing that out. Each figure has been verified for their description and the link in the text, and some have been modified to improve the explanation. Due to the similarities between the different CRs, figures do not always show the main points of the paragraph. Therefore, we have decided to highlight particular points in some figures instead of the common characteristics that are too similar.

Comment 5: The main problem is in the methods section. It does not clearly explain the modeling tools used.

Response 5: As this is a review article, we did not model any proteins. All the proteins of the figures are available online (mostly in the PDB database) and are only shown to illustrate the text. The tools used for each figure have been added to clarify this. About Section 4, we had removed some technical parts to create a supplementary information file according to the suggestion of Reviewer 1 in order to keep only the mechanism and maintain the focus on CR. We have also clarified a few points about the homology modelling steps in Section 4.1.1.

Comment 6: There is no comparison of methods like homology and AI-based predictions.

Response 6: Thank you for your suggestion. We added some comparisons and discussion about the advantages of each method in section 4.1.5 Lines 869-875

Comment 7: It also lacks evaluation metrics for model quality.

Response 7: Thank you for the interesting comment. We have agreed your suggestion, and we have modified the section 4.2 to add explanations on the use of Ramachandran plot and two metrics related to CASP and CAMEO, and that are use a lot, the GDT_TS and lDDT. Lines 887-894

Comment 8: If the authors used structures they visualized or aligned, they need to explain how and with what software.

Response 8: Thank you for the correction. Each figure caption has been updated to be more complete with the methods and software used: Maestro version 14.2.121 (Schrodinger, LLC, Portland, OR).

Comment 9: The discussion and conclusion just repeat earlier points and do not offer new insights into the challenges of predicting structures for different CR families.

Response 9: In this review, we have written Section 6 more as an opening on the utility of models and structures for pharmacophore analysis and other studies on ligands impacting medical and semiochemical research. The added section 5.3 discusses more of the remaining challenges in modeling and structures. To complete this conclusion according to your proposition, we have incorporated a few lines about the impact of IA and data sciences. Lines 1117-1120

Comment 10: They also do not suggest a useful research plan.

Response 10: The conclusion of the review in section 6 suggests using models and structures of CRs in pharmacophore protocols to improve the knowledge on CRs interactions with ligands which could be useful for research in medicine (glucose homeostasis, tumors, etc.) and semiochemistry (insecticides, animal welfare, etc).

Comment 11: The paper does not address the limitations of in silico methods, such as errors in loop regions, membrane placement, or ligand interactions. This makes it incomplete for a high-level journal.

Response 11: Thank you for your suggestions, we have included more discussion about the limitations of in silico method in section 4.1.5. We did not talk about ligand interactions to avoid a tool long article so, we focused only on the structure and address ligand interaction at the end of the review as research subject using models and structures. Lines 1089-1096

In short, while the review has a lot of content, it needs major changes to meet academic standards. It needs more technical detail, clear methods, figures, and a more critical and insightful discussion.

Response: Thank you for all the comments, we have developed many points and improved the overall quality of the text according to your suggestions with the hope that it will meet academic standards now.

Round 2

Reviewer 1 Report

Comments and Suggestions for Authors

The authors have addressed all my concerns. However, the first sentence in the abstract describing semiochemicals is misleading, as this review is about all chemosensensory ligands including odorants, tastants, semoiochemiczls etc.

Author Response

First of all, we would like to thank the reviewers for their works and suggestions on our manuscript. We have addressed all of the remaining queries raised by the reviewers. These modifications are minor and will not affect the framework of the manuscript. The changes were displayed in the revised manuscript in revision mode and highlighted in the PDF file. We have also responded to each of the comments.

The authors have addressed all my concerns. However, the first sentence in the abstract describing semiochemicals is misleading, as this review is about all chemosensensory ligands including odorants, tastants, semoiochemiczls etc.

Thank you for pointing this out, we agree with your comment. Thus, we have changed the first sentence of the abstract to include more chemical cues as in the introduction. Lines 7-9

Reviewer 2 Report

Comments and Suggestions for Authors

The authors have addressed the requested comments; therefore, I recommend accepting the manuscript without further revisions.

Author Response

First of all, we would like to thank the reviewers for their works and suggestions on our manuscript. We have completed the remaining queries raised by the reviewers. These modifications are minor and will not affect the framework of the manuscript. The changes were displayed in the revised manuscript in revision mode and highlighted in the PDF file. We have also responded to each of the comments.

The authors have addressed the requested comments; therefore, I recommend accepting the manuscript without further revisions.

Thank you for your recommendation and approval. We have changed the first sentence of the abstract according to the comment of Reviewer 1.  Lines 7-9
